# Morphoregulatory ADD3 underlies glioblastoma growth and formation of tumor–tumor connections

Carlotta Barelli[1], Flaminia Kaluthantrige Don[1], Raffaele M Iannuzzi[1], Stefania Faletti[1], Ilaria Bertani[1], Isabella Osei[1], Simona Sorrentino[1], Giulia Villa[1], Viktoria Sokolova[1], Alberto Campione[1,2], Matteo R Minotti[2], Giovanni M Sicuri[2], Roberto Stefini[2], Francesco Iorio[1], Nereo Kalebic[1]

**Glioblastoma is a major unmet clinical need characterized by striking inter- and intra-tumoral heterogeneity and a population of glioblastoma stem cells (GSCs), conferring aggressiveness and therapy resistance. GSCs communicate through a network of tumor–tumor connections (TTCs), including nanotubes and microtubes, promoting tumor progression. However, very little is known about the mechanisms underlying TTC formation and overall GSC morphology. As GSCs closely resemble neural progenitor cells during neurodevelopment, we hypothesized that GSCs' morphological features affect tumor progression. We identified GSC morphology as a new layer of tumoral heterogeneity with important consequences on GSC proliferation. Strikingly, we showed that the neurodevelopmental morphoregulator ADD3 is sufficient and necessary for maintaining proper GSC morphology, TTC abundance, cell cycle progression, and chemoresistance, as well as required for cell survival. Remarkably, both the effects on cell morphology and proliferation depend on the stability of actin cytoskeleton. Hence, cell morphology and its regulators play a key role in tumor progression by mediating cell–cell communication. We thus propose that GSC morphological heterogeneity holds the potential to identify new therapeutic targets and diagnostic markers.**

## Introduction

Glioblastoma (GBM) is the most aggressive and common form of primary brain malignancy in adults and an unmet clinical need (Tran & Rosenthal, 2010). Its high chance of relapse is largely due to its striking inter- and intra-tumoral heterogeneity along with its infiltration into the healthy brain parenchyma (Petrecca et al, 2013; Spiteri et al, 2019; Garofano et al, 2021). Cellular interactions between GBM cells and the microenvironment were shown to be important to maintain the aggressive character of the tumor (Osswald et al, 2016; Pinto et al, 2020; Yabo et al, 2022). GBM cells form intercellular networks via two main types of tumor–tumor connections (TTCs): tunneling nanotubes (TNTs) and tumor microtubes (TMs) (Pinto et al, 2020; Zurzolo, 2021; Venkataramani et al, 2022a). Through these connections, cancer cells form a multicellular network, which affects GBM proliferation (Osswald et al, 2015; Ratliff et al, 2023), invasion (Osswald et al, 2015; Lu et al, 2017, 2019; Venkataramani et al, 2022b), and therapy resistance (Osswald et al, 2015; Weil et al, 2017; Hekmatshoar et al, 2018; Kolba et al, 2019). Despite such important role of cellular protrusions in GBM, little is known about the morphological heterogeneity of GBM cells, the molecules underlying it, and its role in cell proliferation.

Given the striking similarities between neurodevelopment and GBM progression (Azzarelli et al, 2018; Curry & Glasgow, 2021), neural progenitor cells could offer key insights into the molecular and cellular underpinnings of GBM cell morphology and its role in cancer progression. Moreover, a specific type of GBM cells, known as glioblastoma stem cells (GSCs), conferring aggressiveness and therapy resistance to the tumor (Azzarelli et al, 2018; Neftel et al, 2019), shows remarkable similarities to a population of neural progenitor cells called basal or outer radial glia (bRG or oRG), a key cell type underlying fetal development of the human cortex (Fietz et al, 2010; Hansen et al, 2010; Reillo et al, 2011). Not only do GSCs show transcriptomic signatures of bRG (Bhaduri et al, 2020; Couturier et al, 2020), but they also undergo a characteristic type of cell movement, called mitotic somal translocation (MST), previously reported only in fetal bRG (Hansen et al, 2010; LaMonica et al, 2013; Bhaduri et al, 2020). In bRG, cell morphology was shown to have an important role in underlying cell proliferation, migration, and MST (Taverna et al, 2014; Ostrem et al, 2017; Molnar et al, 2019; Kalebic & Huttner, 2020; Del-Valle-Anton and Borrell, 2022). In fact, different bRG morphotypes were identified (Betizeau et al, 2013; Reillo et al, 2017; Kalebic et al, 2019) and increased morphological complexity has been linked to a greater proliferative potential (Kalebic et al, 2019). Considering such role of cell morphology in neurodevelopment and the presence of tumor microtubes in GBM, we hypothesized that morphological complexity affects GBM progression.

Here, we identified adducin-γ (ADD3), an actin-associated protein (Kiang & Leung, 2018) known to control bRG morphology and proliferation (Kalebic et al, 2019), as a putative master morphoregulator of

---

[1]Human Technopole, Milan, Italy   [2]Ospedale Nuovo di Legnano, Legnano, Italy

Correspondence: nereo.kalebic@fht.org

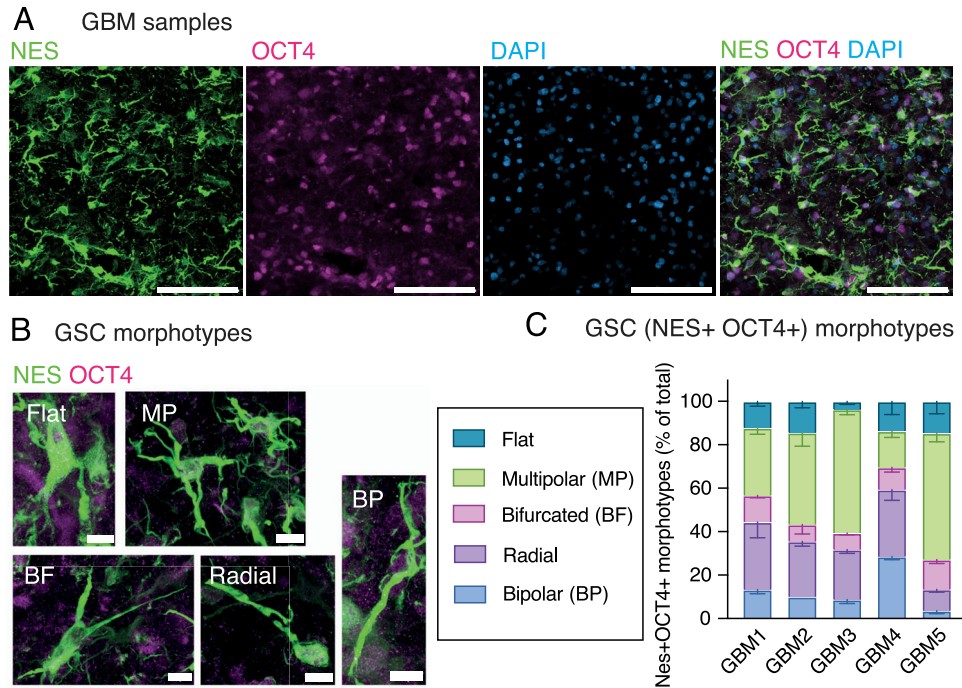

**Figure 1. Glioblastoma stem cells (GSCs) in patient samples exhibit morphological heterogeneity.**
GBM patient samples were immunostained for markers of stemness followed by the analysis of cell morphology. **(A, B)** Immunofluorescence (IF) for nestin (green) and OCT4 (magenta) and DAPI staining (blue), max intensity projection of 25 planes. **(A)** Overview image. **(B)** Five different GSC morphotypes. **(C)** Quantitative analysis of the distribution of GSC morphotypes. Error bars, SEM; n = 3 fields of view. **(A, B)** Scale bars: 200 μm (A); 10 μm (B).

GSCs. We next investigated the morphological heterogeneity of GSCs in patient samples and different GBM cell lines and found that they exist in four morphoclasses, similar to neural progenitors in the developing brain. We demonstrated that ADD3 regulates the morphology of GSCs by inducing their elongation, branching, and the formation of TTCs. We further showed that the effect of ADD3 on cell morphology is necessary for cell survival and correct cell cycle progression. Hence, we described cell morphology as a new layer of heterogeneity in GBM and identified morphoregulatory proteins as potential targets to tackle GBM progression.

## Results

### GSCs exhibit morphological heterogeneity similar to neural progenitors during cortical development

To explore the putative GSC morphological heterogeneity in patient samples, we identified GSCs by immunofluorescence for markers OCT4, a transcription factor that labels pluripotent stem cells, and nestin, an intermediate filament protein that marks neural stem cells and enables visualization of the cell shape (Fig 1A). This allowed us to identify five different morphotypes as follows: flat, multipolar, bifurcated, radial, and bipolar (Fig 1B). We observed all five morphotypes in all the examined samples, albeit with different relative abundance (Fig 1C), suggesting that GSC morphology might also contribute to the prominent heterogeneity of GBM.

Morphologically, these cells were reminiscent of neural progenitor cells during cortical development (Kalebic & Huttner, 2020). Specifically, radial, bifurcated, and bipolar cells morphologically resemble morphotypes of bRG, whereas multipolar cells resemble

multipolar basal progenitors (Kalebic et al, 2019). Instead, flat GSCs do not seem to have a corresponding developmental morphotype and likely arise during tumorigenesis. This suggests that in addition to the molecular and cell behavioral features (Bhaduri et al, 2020), GSCs also recapitulate the morphological features of embryonic neural progenitors, particularly bRG.

### Identification of morphoregulatory *adducin-γ* (ADD3) in GBM

Considering the resemblance between GSCs and bRG, we first sought to identify genes that might govern the GSC morphology by mining datasets of morphoregulators in fetal bRG. We combined previously published transcriptional (Fietz et al, 2012) and proteomic (Kalebic et al, 2019) analyses and identified 45 morphoregulatory genes whose expression is enriched in bRG versus other cell types of the developing brain. We next intersected this list with a published list of genes expressed in GBM (Bhaduri et al, 2020) (Fig 2A). Among the 30 identified genes, the adducin family was prominently present (Fisher's exact test, $P = 3.34 \times 10^{-9}$) with all its three members (Fig 2B and B′).

Adducins are morphoregulatory proteins involved in the assembly of the actin–spectrin network and are implicated in the growth of cell protrusions, in membrane trafficking, and in providing mechanical stability to the plasma membrane (Baines, 2010; Lou et al, 2013; Kiang & Leung, 2018; Kiang et al, 2020). Taking advantage of data from the Cell Model Passports (van der Meer et al, 2019) and the Cancer Dependency Maps (Tsherniak et al, 2017; Behan et al, 2019; Pacini et al, 2021), we excluded genes that were not expressed at the basal level in a panel of commercially available and multi-omically characterized GBM cell lines (Fig 2B) and that are core-fitness essential genes (Vinceti et al, 2021) (Fig S1A) shortlisting a set of 15 candidate genes (Fig 2C).

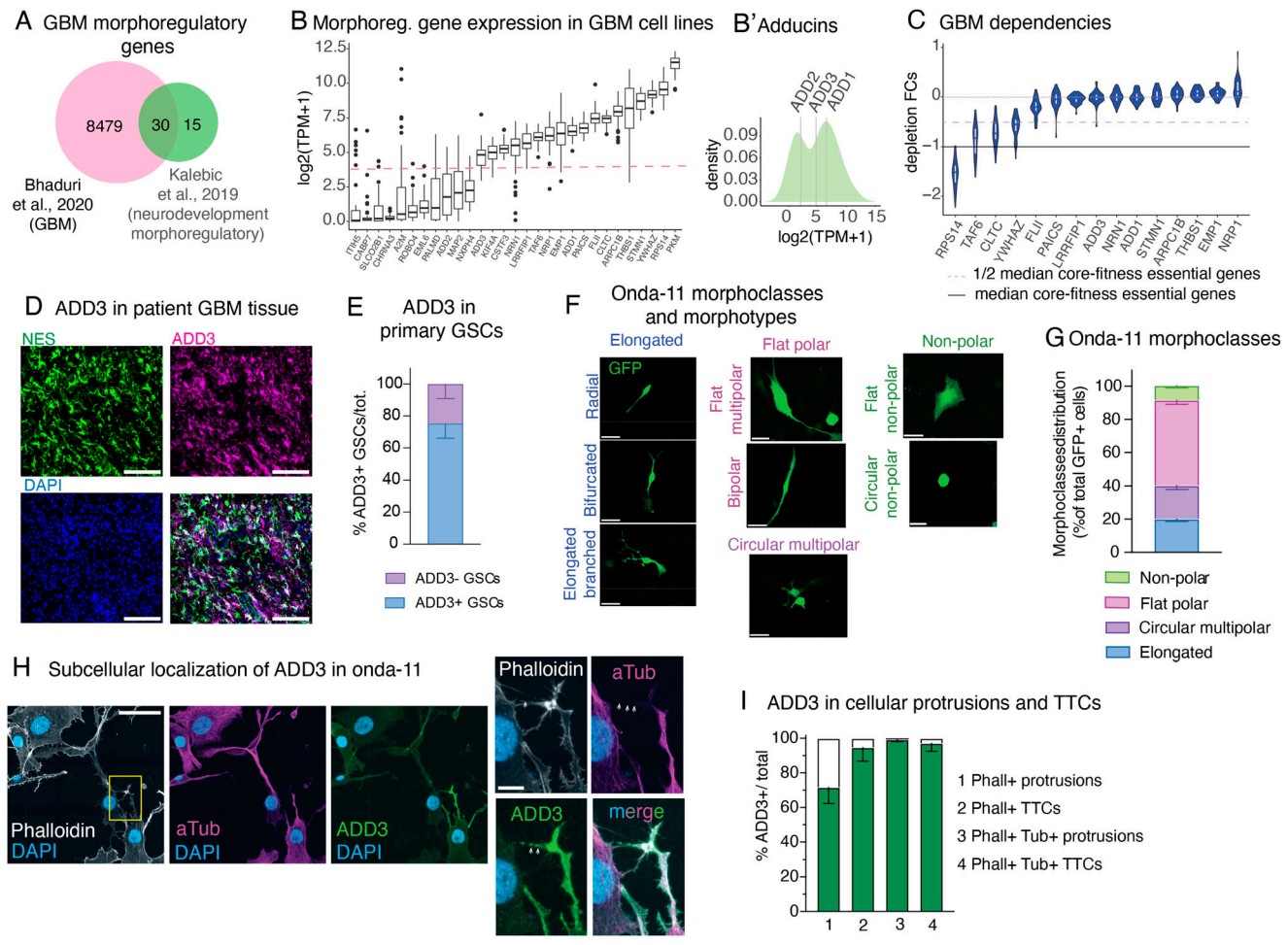

**Figure 2. Onda-11 glioblastoma stem cells (GSCs) show morphological heterogeneity and are dependent on ADD3, a neurodevelopmental morphoregulator localized in GBM cell protrusions and tumor–tumor connections.**

**(A, B, C)** Computational identification of ADD3 as a neurodevelopmental morphoregulator with a putative role in GBM progression. **(A, B, C)** Data are from Kalebic et al (2019) and Bhaduri et al (2020) (A) and Broad DepMap 22Q2 version and Sanger Cell Model Passports (B, C). **(A)** Intersection between a list of 8509 differentially expressed genes in primary GBM tumors and 45 neurodevelopmental morphoregulatory genes, resulting in 30 shared genes. **(B)** Log$_2$(TPM + 1) expression levels of the resulted gene list (29/30) averaged across 48 annotated GBM cell lines from showing bimodal distribution (dashed line). **(B')** Density plot of the average expression levels of the adducin family of genes indicating the estimated density with superimposed average expression levels of adducins. **(B, C)** Dependency of GBM cell lines (depletion fold change [FC] distribution upon CRISPR/Cas9 targeting) on the 15 highly expressed non-core-fitness genes from panel (B). **(D, E)** ADD3 is expressed by GSCs in the primary GBM tissue. **(D)** IF staining of the patient-derived GBM tissue for nestin (green) and ADD3 (magenta) along with DAPI staining (blue), max intensity projection (MIP) of 12 planes. Scale bar: 100 $\mu$m. **(E)** Quantification of the expression of ADD3 in primary GSCs (defined as nestin+, SOX2+). Error bars, SEM; n = 4 independent patient samples. **(F)** Onda-11 GSCs were transfected with GFP, and their cell morphology was analyzed 72 h later. Images are MIPs of 12 planes. Four different morphoclasses listed at the top of the images (elongated, circular multipolar, flat polar, and nonpolar) are further divided into eight morphotypes annotated on the left of the images (radial, bifurcated, elongated branched, circular multipolar, flat multipolar, bipolar, flat nonpolar, or circular nonpolar). Scale bars: 10 $\mu$m. **(G)** Analysis of Onda-11 morphology using GFP signal, 72 h after transfection, showing their morphological heterogeneity. Distribution of the four morphoclasses is shown (see also Fig S1J). Data are the mean of eight independent transfections. Error bars, SEM. **(H, I)** ADD3 is expressed in cellular protrusions and tumor–tumor connections of Onda-11 GSCs. **(H)** IF staining for actin (phalloidin, white), microtubules (alpha-tubulin, magenta), and ADD3 (green) along with DAPI staining (blue). Images are MIP of 12 planes. Scale bars: 50 $\mu$m (left); 10 $\mu$m (right). **(I)** Quantification of the expression of ADD3 in Onda-11 GSC protrusions and microtubes. Error bar, SD; n = 3 independent cell cultures.

Of the 3 adducins, 2 (ADD1 and ADD3) were in this list, with adducin-γ (ADD3) showing a strong and seemingly context-specific essentiality (Fig 2C). Although to our knowledge ADD1 has not been associated with GBM, ADD3 has been reported to both promote and reduce GBM growth and invasiveness (Rani et al, 2013; Kiang et al, 2020). Furthermore, ADD3 has been associated with temozolomide (TMZ) resistance (Poon et al, 2015), glioma progression (Rani et al, 2013; van den Boom et al, 2003), and reduced glioma cell motility (Mariani et al, 2001). Strikingly, we have previously shown that ADD3 is required for the correct morphology of human basal progenitors and that its depletion results in a reduction of their proliferation (Kalebic et al, 2019).

Given these analyses, we examined the expression pattern of ADD3 in the human primary GBM tissue and found that ADD3 is expressed in all patient samples we analyzed (Fig 2D and E). Immunofluorescence staining revealed that 75% of GSCs identified through nestin and SOX2 were also positive for ADD3. Taken together, this prompted us to examine the role of ADD3 in the regulation of GSC morphology.

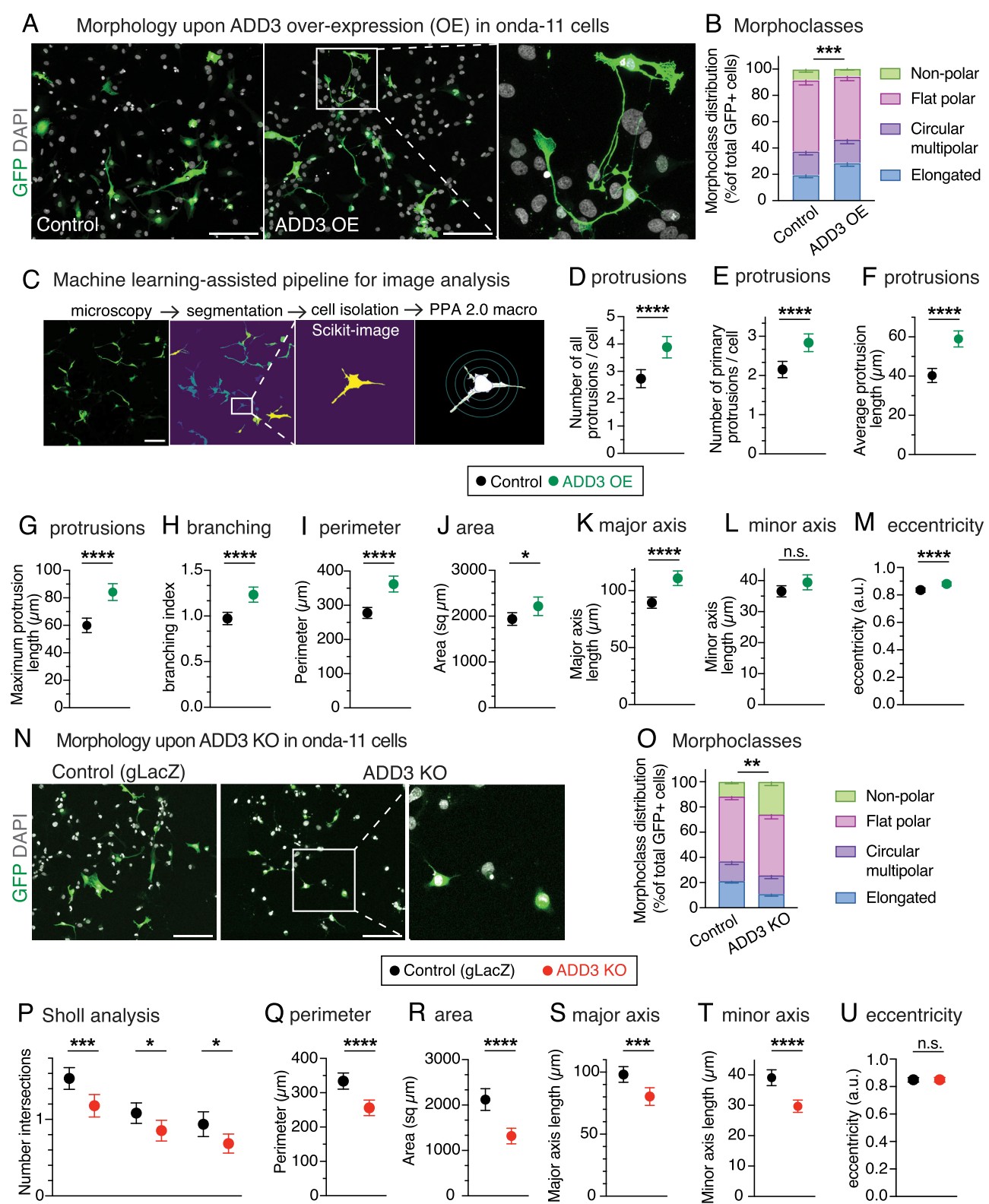

**Figure 3. ADD3 regulates Onda-11 glioblastoma stem cell (GSC) morphology and protrusion number.**
**(A, B, C, D, E, F, G, H, I, J, K, L, M)** ADD3 overexpression promotes cell elongation and protrusion abundance. Onda-11 cells were transfected either with GFP and ADD3-overexpressing plasmids (ADD3 OE) or with a GFP and an empty vector (control), and their morphology was analyzed. **(A)** Representative examples of GFP+ (green) Onda-11 cell morphology in control (left) and ADD3 OE (center). Scale bar: 200 μm. A close-up of elongated cells upon ADD3 OE (right, image width: 250 μm) is shown with the max intensity projection (MIP) of 12 planes. **(B)** Distribution of the four morphoclasses in control and ADD3 OE Onda-11 GSCs. **(C)** Schematics of the pipeline for automated

## Morphological heterogeneity of GSCs and subcellular localization of ADD3

Analysis of Cancer Dependency Map datasets revealed that the GBM cell line Onda-11 exhibits the strongest dependency on ADD3 (scaled depletion fold change upon CRISPR/Cas9 targeting = −0.59, with −1 indicating the median depletion fold change of strongly essential core-fitness genes, such as ribosomal protein genes, Fig S1B). To promote stemness of Onda-11 cells, we maintained them in serum-free culture conditions and confirmed their stem-like features by immunofluorescence for nestin, SOX2, L1CAM, OCT4, GFAP, and CD44 (Fig S1C–I). Importantly, these stemness markers are lost when Onda-11 cells are cultured in serum (Fig S1J). Stemness of Onda-11 GSC was further confirmed through the clonogenic assay in methylcellulose, which revealed that 9.3% of Onda-11 GSCs are able to form clones in stringent conditions at the first and 11.7% at the second serial replating (Fig S1K and L).

Upon transfection with GFP, we examined the morphology of Onda-11 GSCs and found remarkable heterogeneity identifying eight morphotypes (Fig S1M), which we grouped into four principal morphoclasses: nonpolar, flat polar, circular multipolar, and elongated (Fig 2F and G). With the term morphoclass, we refer to a family of morphotypes with the same principal features. Nonpolar cells do not have any type of protrusion, flat polar cells are characterized by a big and flat cell body with some protrusions, circular multipolar cells are small and rounded cells with many short protrusions, and lastly, elongated cells have a long and thin cell body with one or more long and thin protrusions. We further confirmed the existence of the four morphoclasses in another GBM cell line (U-87MG; see Fig S3D).

We examined the subcellular localization of ADD3 in Onda-11 GSCs by confocal microscopy. We observed that ADD3 readily localizes to the proximity of the plasma membrane, to cellular protrusions, and, specifically, to TTCs (Fig 2H). Whereas ADD3 was enriched in protrusions that contained both microtubules and actin, in TTCs it was present irrespectively of whether they contained actin only or actin and microtubules (Fig 2I). Considering the morphological heterogeneity of GSCs and the localization of ADD3 to cellular protrusions, we next sought to examine the potential ability of ADD3 to affect GSC morphology and its role in GBM growth.

## ADD3 is sufficient and required to control the number of protrusions and elongation of GSCs

We transfected Onda-11 GSCs with ADD3-overexpressing (ADD3 OE) and control plasmids along with GFP, to visualize cell shape, and performed a morphological analysis 3 d after transfection (Figs 3A and S2A). ADD3 OE led to an altered distribution of morphoclasses with a marked increase in the proportion of elongated cells at the expense of the other three morphoclasses (Fig 3B). To examine various features of cell morphology in a quantitative manner, we established a machine learning–assisted pipeline for the automatic segmentation and analysis of microscopy images (Fig 3C). Employing this pipeline to examine the effects of ADD3 OE, we observed a striking increase in the number of cellular protrusions (Fig 3D), both primary protrusions that grow directly from the cell body (Fig 3E), and all protrusions, which include also secondary and other higher order protrusions, compared with the control. This was accompanied by an increase in both the average and the maximum length of cell protrusions (Fig 3F and G), which was confirmed also by the Scholl analysis (Fig S2B), and by an increase in protrusion branching (Fig 3H). Together, this suggests that ADD3 promotes both the formation and the growth of new protrusions. Besides, such an increase in cellular protrusions also enlarged cell perimeter and area (Fig 3I and J). Finally, the overall shape of ADD3-overexpressing cells became more elongated, as their major axis was significantly longer than in the control cells, whereas the length of the minor axis was not affected (Fig 3K and L). Accordingly, cell eccentricity was increased, indicating a more elliptic and elongated shape as opposed to circular (Fig 3M).

We next examined whether ADD3 was required to maintain the correct Onda-11 morphology. We performed a CRISPR/Cas9-mediated knockout (KO) of ADD3 and confirmed its efficiency by both immunoblot and immunofluorescence 3 d after transfection (Fig S2C and D). Inspection of the Onda-11 GSC morphology upon ADD3 KO showed altered distribution of morphoclasses with an apparent reduction in the proportion of the elongated cells and a relative increase in the nonpolar cells (Fig 3N and O). Consistent with this and opposite to the effects of the overexpression, ADD3 KO resulted in the reduction of the number of protrusions, their length, branching index, cell perimeter, and area (Figs 3P–R and S2E–J). This was accompanied by a reduction in both major and minor axis lengths (Fig 3S and T), but did not result in a statistically significant reduction in cell eccentricity (Fig 3U).

We examined whether the above effects of ADD3 on cell morphology are pertinent to other GBM cell lines. We used U87-MG GBM line and H4 neuroglioma line that displayed a low expression of stemness markers similar to Onda 11 grown in serum but differently from Onda 11 GSC (Fig S3A and B) and performed ADD3 KO (Fig S3C, D, G, and H). Whereas U87-MG showed strong morphological heterogeneity (Fig S3C, E, and F), which was comparable to Onda-11, H4 cells exhibited rather uniform morphologies (Fig S3G and I). Accordingly,

---

cell segmentation and morphological analysis of cells. GFP+ cells from confocal microscopy images (MIPs of 25 planes) are segmented in CellPose, and single cells are isolated to carry out morphological analysis in Python and Fiji using PPA 2.0 macro. Scale bars: 200 μm. Close-up, 177 μm wide. **(C, D, E, F, G, H, I, J, K, L, M)** Number of total (D) and primary (E) cell protrusions, average (F) and maximum (G) protrusion length, branching index (H), perimeter (I), area (J), major (K) and minor (L) axis length, and eccentricity (M) upon ADD3 OE versus control, calculated as described in (C). **(N, O, P, Q, R, S, T, U)** ADD3 KO reduces protrusion abundance and induces cell shrinkage. Onda-11 cells were transfected either with an ADD3 KO plasmid or with a gLacZ KO plasmid as a control, and their morphology was analyzed. **(N)** Representative examples of GFP+ (green) Onda-11 cell morphology in control (left) and ADD3 KO (center). Scale bar: 200 μm. A close-up of cells upon ADD3 KO (right, image width: 300 μm) is shown with the MIP of 12 planes. **(O)** Distribution of the four morphoclasses in control and ADD3 KO Onda-11 GSCs. **(C, P, Q, R, S, T, U)** Sholl analysis (P), perimeter (Q), area (R), major (S) and minor (T) axis length, and eccentricity (U) upon ADD3 KO versus control, calculated as described in (C). **(B, D, E, F, G, H, I, J, K, L, M, O, P, Q, R, S, T, U)** Data are the mean of four (B, D, E, F, G, H, I, J, K, L, M) and eight (O, P, Q, R, S, T, U) independent transfections. **(D, E, F, G, H, I, J, K, L, M, P, Q, R, S, T, U)** Total number of cells scored: 328 (ADD3 OE) and 397 (control) (D, E, F, G, H, I, J, K, L, M); 317 (KO and control) (P, Q, R, S, T, U). **(B, D, E, F, G, H, I, J, K, L, M, O, P, Q, R, S, T, U)** Error bars, SEM (B, O), 95% CI (D, E, F, G, H, I, J, K, L, M, P, Q, R, S, T, U); *$P < 0.05$; **$P < 0.01$; ***$P < 0.001$; ****$P < 0.0001$; n.s., not statistically significant; two-way ANOVA with Sidak's post hoc tests (B, O), and t test (D, E, F, G, H, I, J, K, L, M, P, Q, R, S, T, U).

KO of ADD3 resulted in a change in morphotype distribution in U87-MG (Fig S3F), but not H4 cells (Fig S3I). Similar to Onda-11, ADD3 KO in U87-MG cells resulted in an increase in nonpolar cells at the expense of elongated ones (Fig S3E and F). This suggests that the effects of ADD3 are pertinent to other GBM cell lines, particularly those that exhibit a heterogeneous cell morphology.

Taken together, these analyses show that ADD3 is both sufficient and required to maintain correct cell morphology, including the correct number and length of cellular protrusions, their branching, cell size, and elongation. Instead, ADD3 is sufficient to increase cell eccentricity, whereas its KO resulted in cell shrinkage without modifying the eccentricity.

### ADD3 promotes morphological transitions during interphase

Considering the above change in the distribution of morphoclasses, we sought to examine potential transitions between GSC morphoclasses using time-lapse microscopy. Two days after transfection with GFP and ADD3 or control plasmids, Onda-11 GSCs were imaged for 60 h. We first focused on the morphological dynamics during interphase and observed that Onda-11 GSCs only rarely undergo a transition between morphoclasses (Fig 4A and B and Video 1). In fact, only the nonpolar cells exhibited morphological dynamics in the interphase (Figs 4C and S4A). ADD3 OE was, however, able to promote such morphoclass transitions (Fig 4A and B) with an increase in transitions into elongated cells (Fig 4A and C and Video 2 and compare with Video 1).

As mitosis involves characteristic morphological changes, we specifically examined the inheritance of the mother cell morphology upon the cell division. During the live imaging, around 40% of cells underwent mitosis (Fig S4B). In control, the mother cell morphology was generally inherited by both daughter cells (Figs 4D and E and S4C). Among the four morphoclasses, nonpolar cells again displayed the greatest number of transitions (Fig 4E and Video 3). Differently to what observed in the interphase (Fig 4B), ADD3 OE was not able to alter the frequency of morphoclass transitions in mitosis (Fig S4C and Video 4). However, upon ADD3 OE, we observed (1) morphological transitions of the progeny of flat polar dividing cells (Figs 4D and E and S4D) and (2) a subtle increase in the elongated progeny of the dividing cells (Fig 4E and Video 5). Both effects were similar to what described above for the interphase. Interestingly, in a subset of elongated cells, we observed an MST-like behavior, which, however, did not seem to be regulated by the overexpression of ADD3 (Fig S4E).

Taken together, these data show that the morphoclass identity is largely conserved in the interphase and in relation to mitosis. The morphological heterogeneity instead seems to be principally generated by the morphological dynamics of nonpolar cells in both the interphase and mitosis. ADD3 overexpression led to an increase in transitions from all morphoclasses into elongated cells, which is consistent with the increase in the proportion of elongated cells described above (Fig 3). Finally, although this effect was mild in mitosis, it led to a marked increase in elongated cells during the interphase.

### ADD3 controls Onda-11 GSC proliferation and survival

Given the (1) effects of ADD3 on GSC morphology (Fig 3) and the previous data showing that (2) ADD3 underlies progenitor

morphology and proliferation during cortical development (Kalebic et al, 2019), we sought to examine the putative effects of ADD3 on the proliferation of Onda-11 GSCs. We first examined the expression pattern of Ki67, a marker of cell proliferation, and categorized cells in three phases of the cell cycle (Fig S5A). Upon ADD3 OE, we detected a relative increase in the proportion of cells in G0 and early G1 phases (Fig 5A and B). This led to a marked reduction in the proportion of cells in the late G2 and M phase, which was confirmed also by immunostaining for a mitotic marker, phospho-vimentin (pVim, Fig S5B and C). Investigating the Ki67 expression pattern across the four morphoclasses revealed the strongest effect in circular multipolar cells and a less prominent one in elongated and nonpolar cells (Fig 5C). Upon EdU treatment of Onda-11 GSCs, we detected no difference in the proportion of cells in the S phase (Fig S5D and E), suggesting that the principal effects of ADD3 OE on GSC proliferation are related to G0/early G1 phases and mitosis.

We next examined the effects of the ADD3 KO on Onda-11 proliferation. Consistent with the above, the KO resulted in the opposite phenotypes compared with the OE. The proportions of cells in both G0/early G1 and late G1/S/early G2 phases were reduced, as revealed by both Ki67 expression pattern and EdU treatment (Figs 5D–F and S6A). We further detected an increase in the proportion of cells in G2/M (Fig 5E), but no specific increase in mitotic pVim+ cells (Fig S6B–D).

Finally, we examined whether the above effects of ADD3 on cell proliferation are pertinent to U87-MG glioblastoma and H4 neuroglioma cell lines. Similar to the effects on cell morphology (Fig S3), ADD3 KO only affected the proliferation of U87-MG cells (Fig S7A–C), but not H4 cells (Fig S7D–F). Taken together, ADD3 enables correct Onda-11 proliferation and this effect is relevant also to other GBM cell lines that show morphological heterogeneity.

Considering the dependence of Onda-11 on ADD3 (Fig S1B), we examined the apoptosis of KO cells by immunofluorescence for cleaved caspase-3 (Fig S6B) and detected a marked increase in cell death compared with the control (Fig 5G). Strikingly, this effect was not specific to transfected cells, but we detected a twofold increase in apoptosis also in the surrounding cells (Fig 5H). Hence, ADD3 is required for the survival of Onda-11 GSCs in both cell-autonomous and nonautonomous manners. Such effects on both the targeted and the neighboring cells prompted us to examine the effects (1) on the Onda-11 molecular signature after ADD3 manipulation and (2) on intercellular connections mediating communication between GSCs.

### Cell-autonomous effects of ADD3 overexpression

To elucidate the cell-autonomous effects of ADD3 OE, we performed a bulk RNA sequencing of GFP+ FACS-sorted cells co-transfected with ADD3 or control plasmids. The differential expression analysis revealed 10 up-regulated and 7 down-regulated genes upon ADD3 OE (Fig 6A). We demonstrated that the genes differentially expressed upon ADD3 OE are indeed exhibiting an expression pattern correlated with ADD3 also at the basal level in other GBM cell lines; that is, the up-regulated genes are correlated, whereas

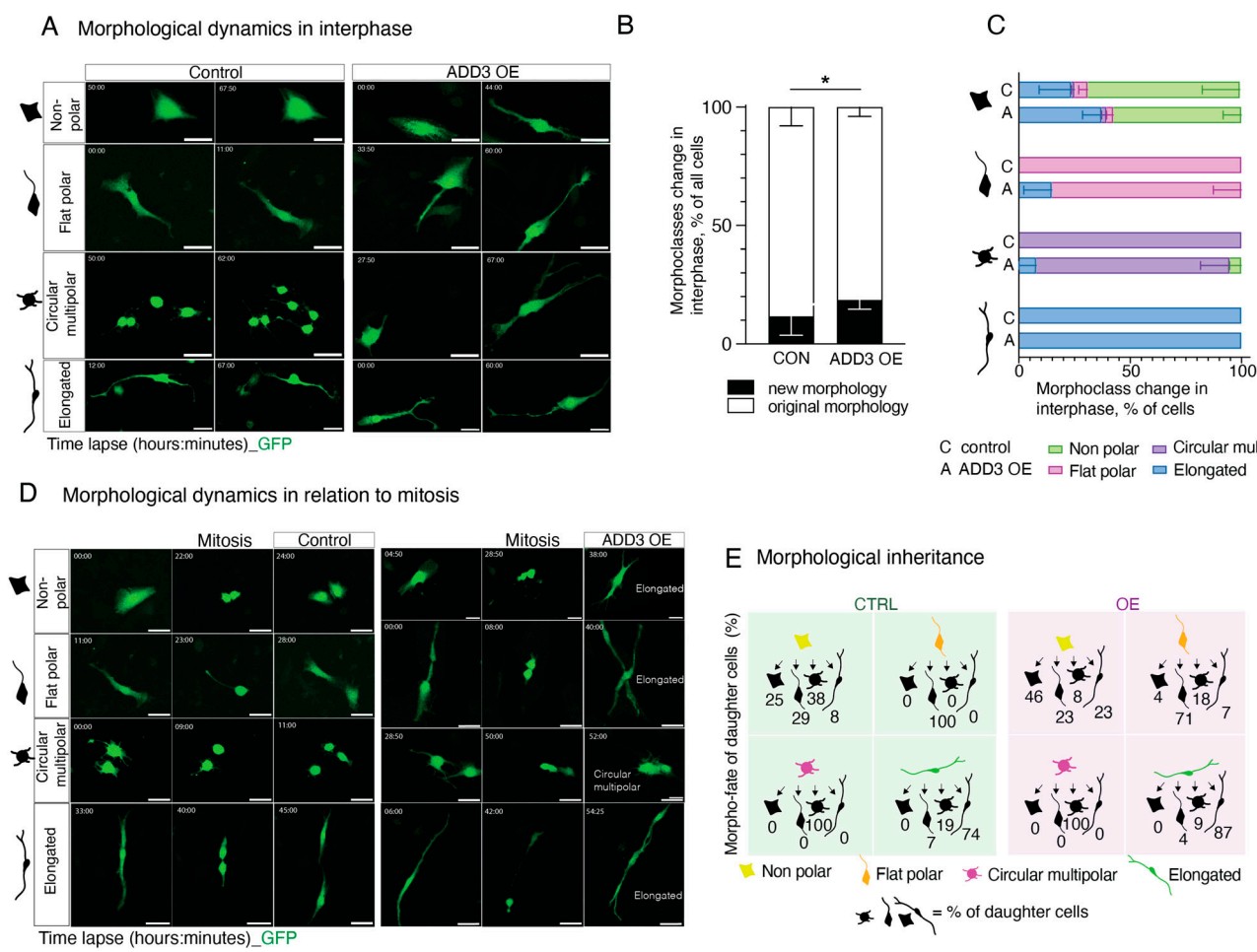

**Figure 4.  ADD3 promotes morphological transitions in the interphase.**
**(A, B, C, D, E)** Onda-11 cells were transfected either with GFP- and ADD3-overexpressing plasmids (ADD3 OE) or with a GFP and an empty vector (control), and their morphological dynamics were analyzed by live imaging in the interphase (A, B, C) and in relation to mitosis (D, E). **(A)** Examples of the morphological dynamics in the interphase of the four morphoclasses upon ADD3 OE versus control. Note the increased elongation of the ADD3 OE cells. The time lapse is indicated in the upper left corner of the images. Scale bars: 50 $\mu m$. **(B)** Quantification of morphological changes in the interphase. Note the increase in acquisition of new morphology upon ADD3 OE. Data are the mean of four independent transfections. Error bars, SD; *$P < 0.05$; two-way ANOVA with the Bonferroni post hoc tests. **(C)** Quantification of morphological transitions in the interphase for each morphoclass. Data are the mean of three independent transfections. Error bars, SEM. **(D)** Examples of the morphological dynamics in relation to mitosis of the four morphoclasses showing the time point pre- (left), during (middle), and post-mitosis (right) upon ADD3 OE versus control. The time lapse is indicated in the upper left corner of the images. Scale bars: 50 $\mu m$. **(E)** Schematic representation of the morphological inheritance shown as the percentage of morphoclass progeny for each mother morphotype. A number of mother cells (control, ADD3 OE) are as follows: nonpolar (12, 13), flat polar (16, 16), circular multipolar (30, 5), and elongated (15, 29). Data are from four independent transfections. See also Video 1, Video 2, Video 3, Video 4, and Video 5.

down-regulated genes are anticorrelated with ADD3 (Fig S8), thus showing robustness of the ADD3 OE signature.

Consistent with the morphoregulatory role of ADD3 (Fig 3), we detected the increased expression of cancer-associated palmitoyltransferase *SPTLC3* (Gruel et al, 2014) and secreted protein *SLPI*, involved in filopodium formation (Mizutani et al, 2020). Furthermore, in accordance with the effects of ADD3 OE on GSC proliferation (Fig 5A–C), we detected down-regulation of *PLK2*, a key regulator of cell cycle progression, involved in centriole duplication and G1/S transition (Cizmecioglu et al, 2008; Chang et al, 2010).

We next examined whether the effects of ADD3 on cell morphology and proliferation had consequences on cell fate and identity. Because ADD3 induced elongated and branched morphologies

of GSCs (Fig 3) and led to a reduction in cell cycle progression and division (Figs 5A–C and S5C), we examined the stemness of ADD3 OE cells and observed that ADD3 sustained as high level of stemness markers as control GSCs (Fig S9A–I). The ADD3 KO in turn led to a minor, albeit not statistically significant, reduction in some of the stemness markers (Fig S9J–R).

Considering that the same morphological and proliferation-related features are also linked to GBM invasiveness (Bhaduri et al, 2020; Venkataramani et al, 2022b), we generated neurospheres from FACS-sorted GFP+ cells overexpressing ADD3 or control plasmid and examined their infiltration into the surrounding Matrigel. However, within 1 wk, we did not observe any difference in the invasion index between ADD3 OE and control cells (Fig S10).

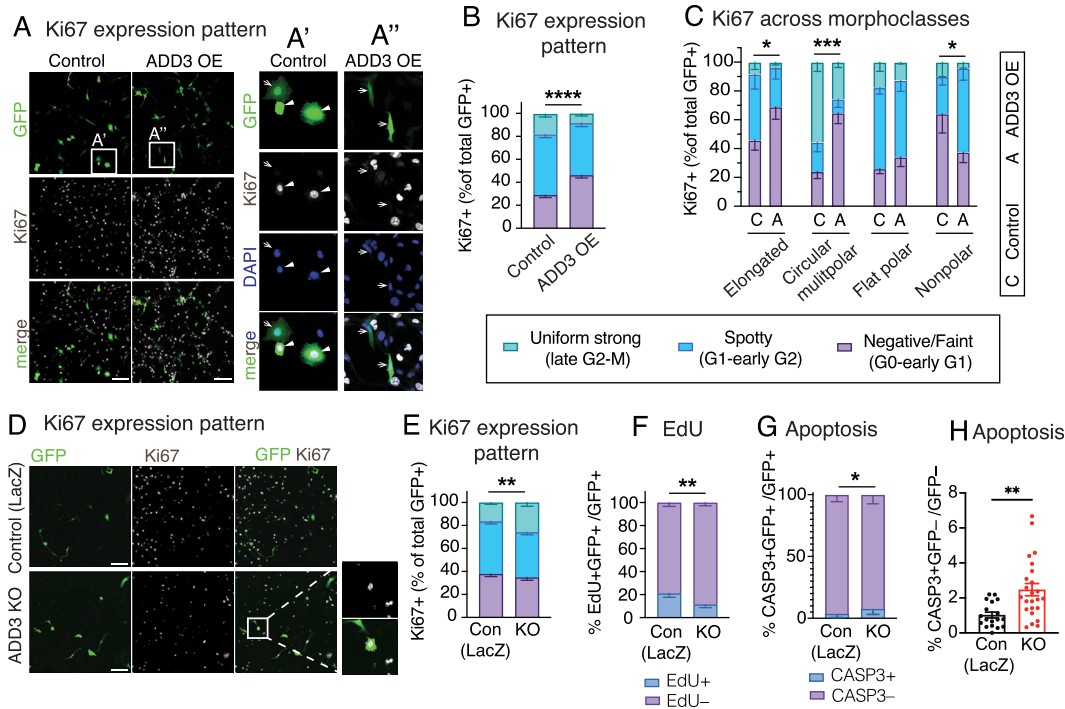

**Figure 5.  ADD3 regulates Onda-11 glioblastoma stem cell (GSC) proliferation and survival.**
**(A, B, C, D, E, F)** Effects of ADD3 OE (A, B, C, D) and KO (E, F) on cell proliferation 72 h after transfection, analyzed by IF for the expression pattern of Ki67, which is indicative of different phases of the cell cycle: uniform strong (late G2/M, green), spotty (G1/early G2, blue), and negative/faint (G0/early G1, violet). Max intensity projections of 12 planes are used to analyze the Ki67 pattern of expression. **(A)** Representative images of IF for Ki67 (white) along with DAPI staining (blue) in GFP+ cells (green). Close-ups of GFP+ control (A') and ADD3 OE cells (A'') are shown. Arrows, Ki67 negative/faint; arrowheads, Ki67 spotty/uniform strong. Note the negative/faint Ki67 expression upon ADD3 OE. **(B, C)** Distribution of the three Ki67 expression patterns in control and ADD3 OE Onda-11 GSCs in the whole population (B) and across morphoclasses (C). **(D)** Representative images of IF for Ki67 (white) along with DAPI staining (blue) in GFP+ cells (green). Close-ups of GFP+ ADD3 KO cells are shown. **(E)** Distribution of the three Ki67 expression patterns in control and ADD3 KO Onda-11 GSCs. Note that ADD3 KO increases the percentage of Onda-11 GSCs in the late G2/M phase. **(F)** Effects of ADD3 KO on cell proliferation 72 h after transfection, analyzed by EdU treatment (4 h) and microscopy. Distribution of EdU+ and EdU- GFP+ Onda-11 GSCs upon ADD3 KO is shown. **(G, H)** Effects of ADD3 KO on cell apoptosis 72 h after transfection, analyzed by IF for cleaved caspase-3 (CASP3) in GFP+-transfected cells (H) and GFP- cells (I). Note the increase in cell apoptosis upon ADD3 KO in both transfected and surrounding cells. **(A, E)** Scale bars: 200 μm (A, E). **(A', A'', E)** Image width: 232 μm (A', A''); 200 μm ((E), insets). **(B, C, F, G, H, I)** Data are from the mean of three (G), four (B, C, F), or eight (H, I) independent transfections. **(B, C, F, G, H)** Error bars, SEM; *$P < 0.05$; **$P < 0.01$; ***$P < 0.001$; ****$P < 0.0001$; n.s., not statistically significant; two-way ANOVA with Sidak's post hoc tests (B, C, F, G, H).

## ADD3 overexpression promotes resistance to Temozolomide

Finally, slowly dividing cells are often associated with therapy resistance (Bao et al, 2006; Chen et al, 2012; Lathia et al, 2015). Interestingly, the expression of ADD3 has been previously linked with a population of cells resistant to Temozolomide (TMZ), the main chemotherapeutic used in GBM treatment (Poon et al, 2015). Furthermore, its expression was also linked to multidrug resistance upon profiling 30 cancer cell lines (Gyorffy et al, 2006). We hence first examined a potential signature of chemoresistance among the genes up-regulated upon ADD3 OE (Fig 6A) and found *CHI3L1* as a key molecule involved in TMZ and radioresistance in GBM cell lines (Akiyama et al, 2014; Shao et al, 2014; Zhao et al, 2020).

To test whether ADD3 OE induces resistance to TMZ-based chemotherapy, we first assessed the effective dose range of TMZ in Onda-11 (Fig S11A) and subsequently performed both acute and chronic treatments of Onda-11 GSCs (Figs 6 and S11B). After co-transfection of Onda-11 GSCs with GFP together with either ADD3 OE or control plasmids, we performed dose–response experiments with various concentrations of TMZ ranging from 200 μM to 600 μM. We calculated the percentage of live GFP+ cells over the total number of cells at d 4 and d 5 after acute TMZ treatment and found that GSCs overexpressing ADD3 were more resistant to the treatment at both time points with all the concentrations tested (Figs 6B and S11B). To then better mimic TMZ therapy in the clinical setting, we performed metronomic administration of 200 μM TMZ every 48 h. Throughout the 7 d of chronic administration, Onda-11 GSCs overexpressing ADD3 had a better viability compared with the control cells (Fig 6C), strongly suggesting that ADD3 promotes resistance of GSCs to chemotherapy.

Taken together, we found that ADD3 (i) promotes protrusion growth and branching, (ii) increases chemoresistance, (iii) reduces cell cycle progression, and (iv) exerts both cell-autonomous and nonautonomous effects on cell survival. Interestingly, chemoresistance and GBM cell proliferation have been strongly associated with a network of TTCs, including TNTs and TMs (Osswald et al, 2015; Weil et al, 2017; Kolba et al, 2019; Wang et al, 2022), so we next sought to examine whether ADD3-related phenotypes are specifically mediated by TTCs.

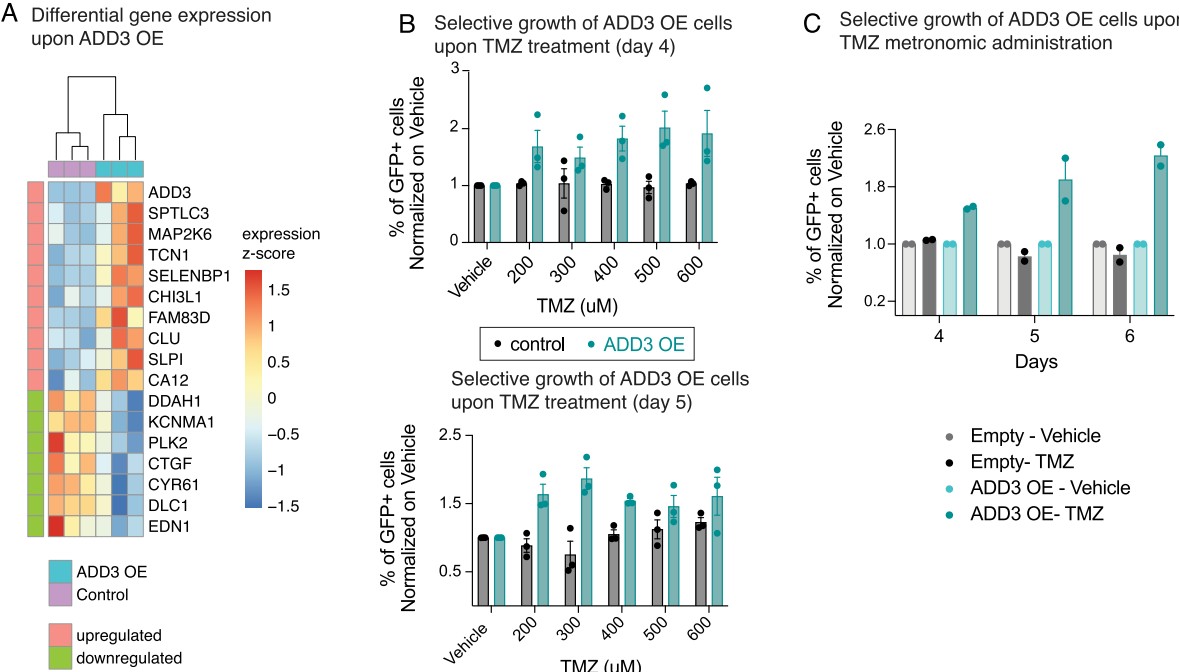

**Figure 6. ADD3 promotes resistance to temozolomide (TMZ).**
**(A)** Differentially expressed genes from contrasting bulk RNA-seq profiles of ADD3 OE Onda-11 versus control, 72 h after transfection. Z-scores of differentially expressed genes (absolute log FC > 0.5 and adjusted $P$ < 0.05) are grouped row-wise according to differential expression sign, with samples hierarchically clustered based on Euclidean similarity. **(B, C)** ADD3 OE promotes resistance to TMZ. **(B)** Quantification of % GFP+ (control or ADD3 OE) cells over total cell number at 4 (upper) and 5 (lower) d after the acute administration of TMZ (200–600 $\mu$M) or vehicle at day 0. Note that upon TMZ treatment, ADD3 OE Onda-11 glioblastoma stem cells have greater survival than control cells. See also Fig S11B for representative images. **(C)** Quantification of % GFP+ (control or ADD3 OE) cells over total cell number upon metronomic administration (every 48 h starting from day 0) of 200 $\mu$M TMZ (darker colors) or vehicle (lighter colors). Note that upon TMZ treatment, ADD3 OE Onda-11 glioblastoma stem cells have greater survival than control cells. **(B, C)** Data are from the mean of three (B) and two (C) independent experiments. **(B, C)** Error bars, SEM. **(B)** Two-way ANOVA, $P$ = 0.02 (day 4); 0.0006 (day 5). **(C)** Three-way ANOVA, $P$ = 0.0001.

## ADD3-induced TTCs are required for the effects of ADD3 on GSC proliferation

To investigate whether ADD3 could affect TTC abundance, we stained Onda-11 GSCs overexpressing ADD3 with phalloidin and $\alpha$-tubulin to detect actin and microtubules, respectively (Fig 7A). We detected doubling of TTCs connecting adjacent cells and containing actin cytoskeleton upon the overexpression of ADD3 (Fig 7B). Using correlative light–electron microscopy, we identified GFP+co-transfected cells and then examined the ultrastructure of ADD3-induced TTCs using cryo-electron tomography (Fig 7C). This showed that such TTCs are strikingly enriched in actin and that no microtubules were observed. Because most of the TTCs were short and thin, they were likely TNTs. Nevertheless, we also observed TMs in control Onda-11 (Fig 2H and I) and upon ADD3 OE (Fig 7A), which was confirmed by IF for connexin-43 (Fig S12).

We thus examined whether intact actin cytoskeleton is required for the maintenance of ADD3-induced protrusions by treating the transfected Onda-11 GSC with cytochalasin D, which causes disruption of actin filaments and inhibits actin polymerization (Fig 7D). Consistent with the above (Fig 7B), DMSO-treated cells exhibited a twofold increase in TTCs upon ADD3 OE (Fig 7E). In contrast, cytochalasin D–treated cells lost all the ADD3-induced TTCs and showed similar levels between the control and OE cells (Fig 7E).

In light of the association between TTCs and cell proliferation (Osswald et al, 2015; Valdebenito et al, 2018; Lu et al, 2019; Venkataramani et al, 2022a; Ratliff et al, 2023) and the ADD3-induced phenotypes on both TTCs and cell cycle progression (Figs 5 and 7A–E), we sought to examine whether the effects of the morphoregulatory ADD3 on cell morphology and TTCs are required for its effects on cell proliferation. We treated control and ADD3 OE Onda-11 GSCs with DMSO and cytochalasin D and examined the expression pattern of Ki67 (Fig 7F) as a key indicator of the effects of ADD3 on cell cycle progression (Fig 5). Our analysis shows that control cells treated with cytochalasin D do not have different cell cycle progression compared with DMSO-treated control cells, suggesting that the stability of the actin cytoskeleton is not required for the normal proliferation of Onda-11 cells. In agreement with what we observed in untreated cells, ADD3 OE GSCs treated with DMSO showed a significant effect on cell proliferation (Fig 7G and compare with Fig 5B), whereas this effect was completely lost upon treatment with cytochalasin D (Fig 7G).

Taken together, these data suggest that ADD3 acts as a key regulator of GSC morphology to induce new actin-rich TTCs, which in turn enable cell–cell contacts and mediate the downstream effects on cell proliferation.

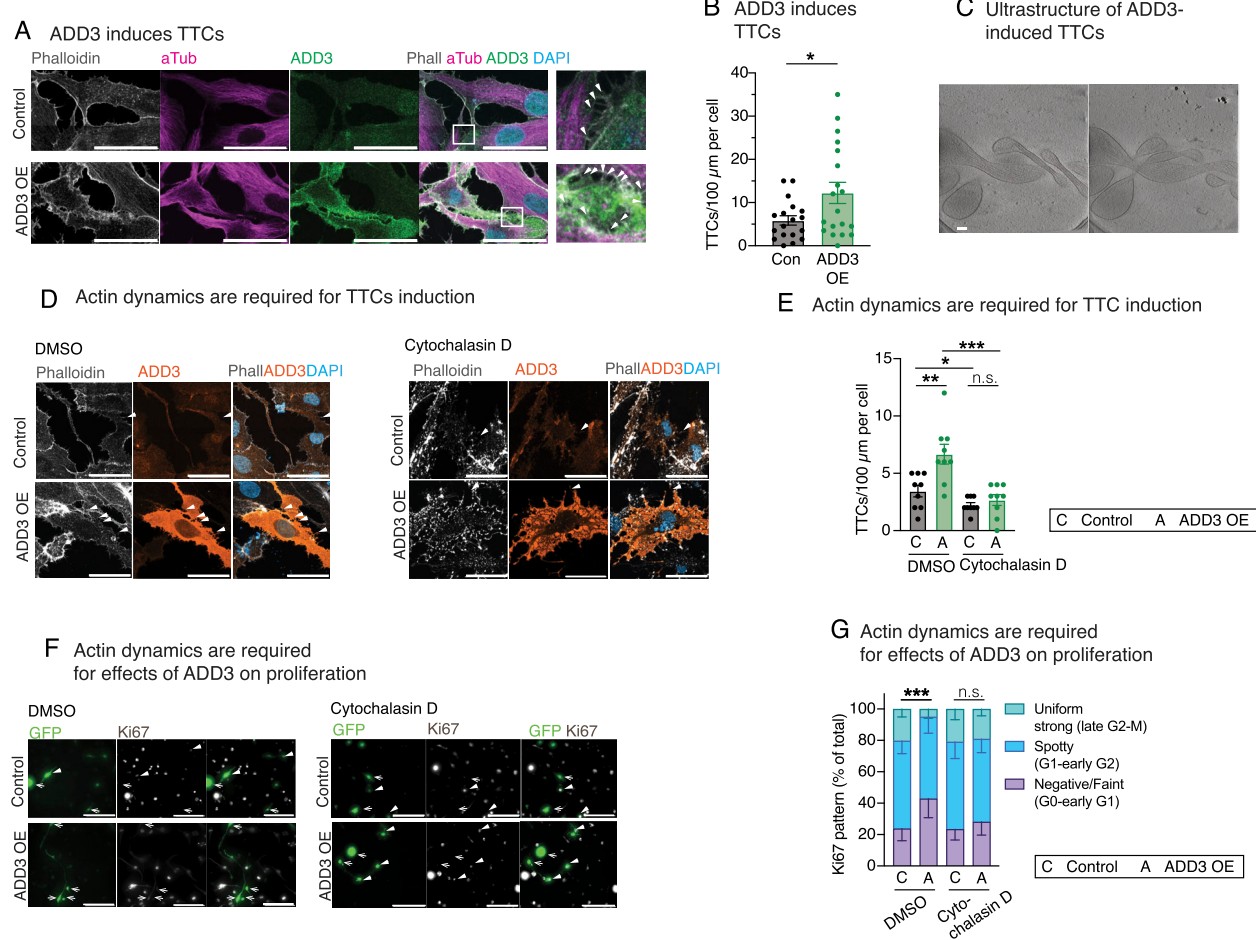

**Figure 7. Effects of ADD3 on cell proliferation are mediated by tumor–tumor connections.**
**(A)** IF for actin (phalloidin, gray), microtubules (α-tubulin, magenta), and ADD3 (green) along with DAPI staining in control (top) and ADD3-overexpressing (OE, bottom) Onda-11 glioblastoma stem cells (GSCs). Arrowheads, microtubes. Images are max intensity projections of 12 planes. Scale bars: 50 μm. **(B)** Quantification of the number of microtubes per 100 μm of cell perimeter, expressed per cell in control and ADD3 OE. **(C)** Two slices of a tomogram showing the ultrastructure of ADD3-induced microtubes, extracted from different Z heights to show intertwining of the protrusions. Note that the microtubes are rich in actin cytoskeleton. Slice thickness: 10 nm; scale bar: 100 nm. **(D, E, F, G)** Actin cytoskeleton is required for both ADD3-mediated induction of microtubes and effects on proliferation. After transfection, Onda-11 GSCs were treated with cytochalasin D (right) at 5 μM concentration for 45 min. **(D)** IF for actin (phalloidin, gray) and ADD3 (orange) along with DAPI staining in control (top) and ADD3 OE (bottom) Onda-11 GSCs treated with 5 μM cytochalasin D (right) and DMSO (left). Arrowheads, microtubes. Images are max intensity projections of 12 planes. Scale bars: 50 μm. **(E)** Quantification of the number of microtubes per 100 μm of cell perimeter, expressed per cell in control and ADD3 OE upon treatment with 5 μM cytochalasin D or DMSO. **(F)** IF for Ki67 (white) in GFP+ (green) Onda-11 GSCs. Arrows, Ki67 negative/faint; arrowheads, Ki67 spotty/uniform strong. Scale bars: 200 μm. **(G)** Distribution of the three Ki67 patterns of expression in control and ADD3 OE Onda-11 GSCs treated with DMSO and 5 μM cytochalasin. **(B, E, G)** Data are the mean of three independent transfections. **(B, E, G)** Error bars, SEM; *P < 0.05; **P < 0.01; ***P < 0.001; n.s., not statistically significant; t test (B, E) and two-way ANOVA with Sidak's post hoc tests (G).

## Discussion

In this study, we identified the GSC morphology as a key player underlying cell proliferation. We further showed that the main driver of this effect is TTCs. There are three aspects of our study that deserve particular discussion: (1) cell morphology is a new layer of GBM heterogeneity; (2) cell–cell connections link GSC morphology with proliferation, chemoresistance, and survival; and (3) ADD3 is a key morphoregulator in GBM.

### Morphology is a new layer of GBM heterogeneity

One of the key reasons for GBM's malignancy is its extraordinary inter- and intra-tumoral heterogeneity. The molecular heterogeneity,

described at genomic, transcriptomic, and epigenetic levels, was shown to underlie a multitude of GBM cell types and states (Sottoriva et al, 2013; Patel et al, 2014; Darmanis et al, 2017; Neftel et al, 2019; Bhaduri et al, 2020; Couturier et al, 2020; Jacob et al, 2020; Chaligne et al, 2021). In fact, it has been suggested that each GBM contains on average 11 different cell types (Bhaduri et al, 2020) that could be grouped into four principal cellular states, which recapitulate distinct neural cell types (Neftel et al, 2019). Notably, GSCs themselves show striking molecular heterogeneity within the same tumor (Bhaduri et al, 2020). However, to link these specific cell types with cellular functions and oncological phenotypes, it is also necessary to study potential GBM heterogeneity at the cell biological level.

We have examined GSC morphology and identified five different morphotypes in primary GBM samples (Fig 1). Importantly, such

morphological heterogeneity was also recapitulated in our 2D GSC model systems, suggesting that basic morphological nature is a cell-intrinsic property. The identified morphotypes bore striking similarity to neural stem cells during cortical development, in particular, bRG (Kalebic & Huttner, 2020). This is consistent with a large body of evidence showing that GBM initiation, maintenance, and progression are controlled by the same signaling pathways and transcription factors that regulate brain development (Azzarelli et al, 2018; Daniel et al, 2018; Curry & Glasgow, 2021). Despite good molecular understanding, the links between neurodevelopment and GBM at the cell biological level remain largely unexplored. It is hence particularly interesting that in GBM, we find morphotypes that are comparable to those that promote proliferation during neurodevelopment (Kalebic et al, 2019).

A key question to answer was whether the morphotypes are stable or transient cellular states. Our live-imaging experiments (Fig 4) suggested that the former is true both within and across cell cycles. It would hence be interesting to examine whether such stable morphotypes correspond to transcriptionally defined cell types. So, how is the morphological heterogeneity generated? Our data showed that the cells of the nonpolar morphoclass are responsible for such heterogeneity, as they are able to generate all the remaining morphoclasses both in mitosis and by morphological transitions in the interphase. This is interesting in the context of the hypothesis on the flexibility of cell polarity, which was proposed to underlie morphological heterogeneity during neurodevelopment (Kalebic & Namba, 2021). However, nonpolar cells are not present among neural progenitors in the interphase (Kalebic & Huttner, 2020), and hence, it is tempting to hypothesize that morphological flexibility might exist in both neurodevelopment and GBM, but it is exhibited by different morphotypes. It further suggests that nonpolar cells might be a prominent feature of brain cancers and that their flexibility of morphology might be linked to the characteristic plasticity among different GBM cell states (Neftel et al, 2019).

### Cell–cell connections link GSC morphology with proliferation, chemoresistance, and survival

To examine whether different morphotypes have distinct cellular functions, we analyzed their proliferation and observed differences in their cell cycle progression (Fig 5). Notably, modifying cell morphology, by the overexpression of ADD3, and thus generating more elongated cells, led to a reduced cell cycle progression. We found that the key morphological feature responsible for the change in proliferation is the actin-based TTCs, including both TNTs and TMs (Figs 7 and S12). Whereas formation of TTCs has been implicated in increased cell proliferation (Osswald et al, 2015; Lu et al, 2019; Joseph et al, 2022), a recent study has shown that TM-rich, interconnected GBM cells have a slower cell cycle compared with the fast-dividing, unconnected cells in the invasion zone (Ratliff et al, 2023), which is in agreement with our data (Figs 5 and 7).

Furthermore, such TTC-rich cells overexpressing ADD3 did not show altered invasive capacity (Fig S10), suggesting that the effects of ADD3 are not specific to invadopodia (Petropoulos et al, 2018) but to other cell protrusions, most notably TTCs. This is in line with recent in vivo studies showing that a different population of GBM cells, which lacks connections to other GBM cells, is the main driver

of brain tumor invasion (Venkataramani et al, 2022b; Ratliff et al, 2023). Taken together, our data suggest that TTC-rich GBM cells overexpressing ADD3 are likely not the invading cells, but rather represent a population of slowly proliferating cells either before the infiltration into the brain parenchyma or after it.

TTC-rich GBM cells have been also associated with increased resistance to chemotherapy (Osswald et al, 2015; Weil et al, 2017; Kolba et al, 2019; Wang et al, 2022). Such cells were shown to be able to change their metabolic profile through a TNT-mediated mito-chondria, vesicle, and protein transfer (Hekmatshoar et al, 2018; Pinto et al, 2021). Indeed, upon ADD3 OE, we also observed an increased resistance to TMZ therapy administered both as an acute dose and as metronomic treatment (Fig 6B and C). This is coupled with the up-regulation of CHI3L1 (Fig 6A), which is involved in chemo- and radioresistance in GBM (Akiyama et al, 2014; Shao et al, 2014; Zhao et al, 2020).

Hence, TTCs appear to be the mediators by which morphology affects GBM progression. Because GSCs' transcriptional heterogeneity is controlled by both intrinsic and extrinsic factors (Prasetyanti & Medema, 2017), we propose that the morphological heterogeneity could also be regulated both cell-autonomously and nonautonomously and that TTCs might play a pivotal role in the latter. In fact, we showed that ADD3, as an intrinsic factor promoting morphological heterogeneity, has a critical role in cancer cell survival both cell-autonomously and nonautonomously (Fig 5G and H). Such effect on surrounding cells might be due to the exchange of specific pro-apoptotic signals through the release of some paracrine or autocrine factors, and extracellular vesicles, or, rather, through the loss of direct cell–cell contact, after the striking reduction in TTCs (Fox & MacFarlane, 2016; Vucetic et al, 2020; Yang et al, 2024).

Taken together, GBM cell morphology mediates intercellular communication and thus has important consequences on tumor cell proliferation, survival, and resistance to therapy. Hence, in GBM, like in other cancers (Alizadeh et al, 2020; Wu et al, 2020; Barker et al, 2022), cell morphology has a strong potential to be used as a diagnostic and prognostic marker, through microscopy-based analysis of the tumor.

### ADD3 as a key morphoregulator in GBM

We have identified ADD3 as a key morphoregulator able to control GBM proliferation, survival, and chemoresistance. We found that ADD3 exerts multiple morphoregulatory functions on GSCs. Notably, it promotes cell elongation and induces various cell protrusions, including TTCs (Figs 3 and 7A–C). Such diverse roles are likely due to its close interaction with actin, a key cytoskeleton component regulating changes in cell shape. Indeed, when the actin cyto-skeleton is disrupted, ADD3 is not able to induce TTCs anymore (Fig 7D and E). The question remains whether ADD3 directly induces new protrusions by remodeling actin in the membrane cytoskeleton or whether it stabilizes existing protrusions by connecting actin fil-aments to the plasma membrane. Previous work on other members of the adducin family seems to favor the latter hypothesis as it has been shown that adducins regulate membrane stability by capping the fast-growing end of actin filaments and connecting spectrin–actin cytoskeleton to membrane proteins (Kuhlman et al, 1996; Li

et al, 1998; Anong et al, 2009; Baines, 2010). Actin capping is essential for filopodium formation that in turn leads to neurite outgrowth (Dent et al, 2007). Accordingly, adducins were shown to stabilize neuronal synapses by controlling spine dynamics (Babic & Zinsmaier, 2011; Bednarek & Caroni, 2011; Pielage et al, 2011). Although adducins operate together as heterodimers that form tetramers (Joshi et al, 1991; Matsuoka et al, 2000), there appears to be a selective contribution of different adducins to specific diseases. For example, only variants in ADD3 have been associated with hereditary cerebral palsy (Kruer et al, 2013; Sanchez Marco et al, 2022), and in the context of GBM, only ADD3 has been implicated in tumor progression and resistance to therapy (Rani et al, 2013; Poon et al, 2015). However, the molecular mechanisms by which the specificity among adducins is achieved remain poorly understood. Taken together, it is tempting to hypothesize that ADD3 stabilizes GSC projections by providing mechanical support. This ultimately can lead to an increase in the number of stable cell protrusions, particularly long TTCs that enable cell–cell communication.

Considering that such a role of ADD3 in actin cytoskeleton is likely true across different types of cellular projections and cell types, it is plausible that its effects are not specific to Onda-11 GSCs, but generally applicable to GBM cells that are elongated and contain protrusions. In support of this, we showed that ADD3 plays an important role in maintaining the cell morphology of U-87MG cells (Fig S3). Furthermore, ADD3 has already been implicated in GBM progression, therapeutic resistance, and cell motility (Kiang et al, 2020; Mariani et al, 2001; Poon et al, 2015; Rani et al, 2013; van den Boom et al, 2003). Nevertheless, the effects of ADD3 on both cell morphology and proliferation are more pronounced in Onda-11 GSC compared with U-87MG (compare Fig 3 with Fig S3 and Fig 5F with Fig S7). We link this to the notion that Onda-11 cells were shown to be strongly dependent on ADD3 in the Cancer DepMap project (Tsherniak et al, 2017; Behan et al, 2019; Pacini et al, 2021), whereas U-87MG were not. Beyond the experimental validation of the findings reported in the Cancer DepMap, our results show that DepMap is an important resource for exploring the function of cancer genes in an appropriate model system. In the future, it would hence be interesting to study the morphoregulatory mechanisms in more complex model systems such as in vivo or in patient-derived organoids.

Finally, ADD3 was previously shown to regulate the morphology of bRG during brain development (Kalebic et al, 2019). Its KO in the human fetal brain tissue led to a reduction in the number of protrusions of neural progenitors, which in turn resulted in a reduction in the proliferative capacity of these cells (Kalebic et al, 2019). This link between cell morphology and proliferation serves as a further example of how neurodevelopment can offer precious insights into brain cancers. It also provides a novel conceptual framework, which allows for the identification and mechanistic characterization of other potential molecular targets to be used in future diagnostic and therapeutic approaches in brain cancers.

# Materials and Methods

The Reagents and Tools table is shown in Table S1.

## Human samples

GBM patient samples were obtained from Ospedale Nuovo di Legnano after informed patient consent. A total of five patient samples from both males (2) and females (3) between 62 and 76 yr old with a diagnosis of grade IV astrocytoma were included. Fresh surgical resections were collected in Hibernate-A Medium (A1247501) containing penicillin–streptomycin 100X (ECB3001D) and amphotericin B 100X (15290018) and transported to Human Technopole on ice for processing. Immediately upon arrival, the tissue was dissected and washed in Hibernate-A Medium to remove cellular debris. After 24 h of fixation in 4% PFA, the tissue was left in 15% and 30% sucrose gradients for 24 h each. After embedding in OCT compound (05-9801), serial sections of 20 $\mu$m were cut at the cryostat and stored at –20°C for immunofluorescence experiments.

## Cell culture

Onda-11 cells were reconditioned to GSCs (Onda-11 GSCs) and grown on laminin (5 $\mu$g/ml, L2020; Sigma-Aldrich)-coated plates in serum-free media (GSC medium) composed of DMEM/F-12 with 15 mM Hepes and L-glutamine (11330057; Thermo Fisher Scientific), P/S, N2 supplement (17502-048; Thermo Fisher Scientific), B27 supplement (17504-044; Thermo Fisher Scientific), EGF (10 $\mu$g/$\mu$l), and FGF2 (10 $\mu$g/$\mu$l). U-87MG and H4 cells were grown in DMEM/F-12 with 15 mM Hepes and L-glutamine, P/S, and 10% FBS (F7524; Sigma-Aldrich).

For transfection, Onda-11 GSCs were plated at a density of 10,000 cells/cm$^2$ and treated with Opti-MEM (31985062; Gibco) containing Lipofectamine Stem Transfection Reagent (LIPO-STEM, STEM008; Thermo Fisher Scientific) and DNA mixture. In each six-well plate, 4.5 $\mu$l of LIPO-STEM, 2.7 $\mu$g of recombinant DNA, and 600 $\mu$l of Opti-MEM were used. The cells were either fixed after 72 h in 4% PFA for 15 min and processed for immunofluorescence (IF), or grown for 48 h, sorted to isolate GFP+ cells, and used for RNA and protein extraction or for neurosphere formation assay. U-87MG cells were transfected with Lonza's 4D-Nucleofector System following the instructions of Lonza P3 Primary Cell 4D-Nucleofector X Kit. Briefly, in each cuvette, 500,000 cells were treated with 100 $\mu$l AMAXA nucleofector solution and 2.5 $\mu$g DNA and then replated and kept in culture for an additional 72–96 h, when they were fixed in 4% PFA. H4 cells were transfected using FuGENE HD (E2311; Promega) following the supplier's reverse transfection protocol. Briefly, for each well of a 24-well plate, 20 $\mu$l of Opti-MEM, 250 ng DNA, and 0.75 $\mu$l FuGENE HD reagent were washed and 30,000 cells were plated. 72–96 h post-transfection, the cells were fixed in 4% PFA.

For the EdU proliferation assay, 72 h post-transfection, Onda-11 GSCs were treated with EdU (Click-iT Plus EdU Alexa Fluor 647) for 4 h, then fixed in 4% PFA for IF and imaging. For the actin cytoskeleton disruption assay, 48 h post-transfection, Onda-11 GSCs were treated with cytochalasin D at a 5 $\mu$M concentration for 45 min, then kept in culture for an additional 4 h, and fixed in 4% PFA for IF experiments.

For the clonogenic methylcellulose assay, 3,000 single Onda-11 GSCs per well were plated into MethoCult methylcellulose-based media (SF H4236; StemCell Technologies) mixed with the GSC

medium (1:1). As the cells grew, they formed colonies upon which we calculated the percentage of single cells that were able to produce colonies. The procedure was repeated for two serial replatings.

The responsiveness of Onda-11 GSCs to TMZ (T2557; Sigma-Aldrich) was assessed by treating them with the TMZ doses ranging from 200 to 1,000 µM. For TMZ resistance experiments, Onda-11 GSCs were co-transfected with GFP and either an empty plasmid or an ADD3 OE plasmid. For the dose–response experiments, the percentage of GFP+ cells was calculated over 5 d after treatment with 200–1,000 µM TMZ. For the metronomic treatment, 200 µM of TMZ was administered every 48 h starting from day 0, mimicking the dose given to patients. Subsequently, the percentage of GFP+ cells was calculated over 6 d of treatment.

### Cell sorting

The cells were sorted 48 h after transfection to isolate GFP+ cells. MoFLO Astrios EQ Cell Sorter, equipped with Summit 6.3.1 software (Beckman Coulter), was used for cell sorting before the neurosphere formation assay, whereas CytoFLEX SRT Cell Sorter, equipped with CytExpert SRT software (Beckman Coulter), was used for cell sorting before RNA and protein extraction. An average sorting rate of 500–1,000 events per second at a sorting pressure of 25 psi (for MoFLO Astrios EQ) or 15 psi (for CytoFLEX SRT) with a 100-µm nozzle was maintained.

### Plasmids

For the overexpression of ADD3, human *ADD3*–encoding cDNA was amplified by PCR, using the forward and reverse primers CAAX_Xhol_Fw and CAAX_BgIII_Rev as reported above, and cloned into the pCAG vector. DNA was purified using QIAquick PCR Purification Kit (28104; QIAGEN), and all DNA plasmids were extracted and purified using the EndoFree Plasmid Maxi kit (12362; QIAGEN) following the manufacturer's instructions.

For CRISPR/Cas9 gene editing of ADD3, two guide RNAs (gRNAs) targeting exons 4 and 6 were cloned into pSpCas9(BB)-2A-GFP (PX458), following the previously published protocol (Ran et al, 2013). For control, a previously published gRNA targeting LacZ was used (Kalebic et al, 2016).

### Immunoblotting

Total cell lysates were prepared in a denaturing buffer (Tris–HCl, pH 7.4, 50 mM, NaCl 150 mM, and 1% SDS). After 15 min of solubilization on a rotating wheel, debris were removed by centrifugation (10,000*g*, 15 min at RT). The protein concentration was determined using the Pierce BCA Protein Assay kit (Thermo Fisher Scientific). Total protein extracts (20 µg) were separated on NuPAGE 4–12% gels (Thermo Fisher Scientific) and blotted onto nitrocellulose membranes (Hybond, GE Healthcare). After blocking with 5% dry milk for 1 h at RT, membranes were incubated overnight at 4°C with antibodies against ADD3 (1:1,000) and actin (1:20,000), washed 3x in TBS-T, and incubated with secondary antibodies (1:10,000) for 1 h at RT, washed 3x, and the signal was detected using ECL (Clarity Western ECL, Bio-Rad) and visualized with a ChemiDoc imaging system (Bio-Rad).

### Neurosphere invasion assay

FACS-sorted GFP+ Onda-11 GSCs were replated in ultra-low attachment 96-well plates starting from 1,000 cells per neurosphere in 200 µl GSC medium. After 3 d in culture, the neurospheres were embedded in 50 µl Matrigel for the invasion assay. Brightfield images were taken every 2 d using EVOS M5000 Imaging System with 4X (0.13 NA) or 10X (0.30 NA) objective. Neurospheres were kept until day 15 when they were fixed in 4% PFA for 20 min at RT.

### Immunofluorescence (IF)

For IF of human patient samples, antigen retrieval was performed by incubating the slides for 45 min with 10 mM citrate buffer, pH 6.0, in a 70°C oven. After washes in PBS, the tissue was permeabilized for 30 min in 0.3% Triton X-100 at RT. After quenching in 0.1 M glycine in PBS at RT for 30 min and blocking in 10% normal donkey serum (017-000-121), 300 mM NaCl, and 0.5% Triton X-100 in PBS at RT for 30 min, primary antibodies anti-nestin (1:500), anti-OCT4 (1:200), anti-SOX2 (1:200), and anti-ADD3 (1:500) were incubated in blocking solution overnight, at 4°C. After washes in PBS, the sections were incubated with secondary antibodies (1:500) and DAPI (1:2,000) in 0.3% Triton X-100 in PBS for 1 h at RT, washed, and mounted on microscopy slides with Mowiol + antifade (81381).

For IF of 2D cell culture, the cells were permeabilized for 30 min in blocking solution containing 5% normal donkey serum and 0.3% Triton X-100 in PBS at RT. Primary antibodies were incubated in blocking solution for 2 h, at RT. The following primary antibodies were used: anti-p-(ser55)-vimentin (1:500), anti-Ki67 (1:500), anti-nestin (1:500), anti-ADD3 (1:500), anti-cleaved caspase-3 (1:300), anti-GFAP (1:1,000), anti-CD44 (1:500), anti-A2B5 (1:300), anti-L1CAM (1:200), anti-OCT4 (1:200), and anti-alpha-tubulin (1:500). After three washes in PBS, the sections were incubated with secondary antibodies (1:500) in 0.3% Triton X-100 in PBS for 30 min at RT, washed again three times in PBS, and imaged within the following 2 wk.

### Light microscopy

Confocal microscopy on fixed cells was performed using a Zeiss LSM 980 point-scanning confocal or Zeiss LSM 980-NLO point-scanning confocal based on Zeiss Observer 7 inverted microscopes. The images were acquired with a PlanApo 10X/0.45 dry or a PlanApo 20X/0.8 dry or a PlanApo 40X/1.4NA oil immersion objective using 405-, 488-, 561-, and 639-nm laser lines. The software used for all acquisitions was Zen Blue 3.7 (Zeiss). Once the parameters of acquisition for control conditions had been defined, they were kept constant for all the samples within the same experiment.

Time-lapse imaging on live Onda-11 GSCs was performed as follows. 48 h after transfection, the sample was placed under a Zeiss LSM 980 point-scanning confocal with a PlanApo 20X/0.8 dry objective and imaged for ~60 h. Z-stacks of 18–20 µm range were taken with a Z-step of 1 µm and an interval time of 30–40 min.

For the TMZ resistance and clonogenic assays, fluorescence (for TMZ) or brightfield (for the clonogenic assay) images were taken using EVOS M5000 Imaging System with a 4X (0.13 NA) or 10X (0.30 NA) objective.

## Correlative light–electron microscopy and cryo-electron tomography

Quantifoil Gold Grids (R 2/2, Au, 200 mesh; Quantifoil) were plasma-cleaned with a hydrogen and oxygen mix (20:80) for 15 s with Gatan Solarus II and then washed for 1 h with 100% EtOH. The grids were then coated with 5 $\mu$g/ml laminin (for 1 h at 37°C), and around 25,000 Onda-11 GSCs were seeded per grid. 16 h later, the grids were plunged with a Leica EM GP2 plunger. During plunging, a drop of 3 $\mu$l BSA-coated 10-nm fiducial gold markers (Aurion) was applied on the EM grids for 2.5 s. Grids were stored in liquid nitrogen until acquisition.

Subsequently, cryo-fluorescence imaging was performed on a Leica Thunder Cryo-CLEM system using the Navigator module of Leica LAS X software. Grids were focus-mapped using built-in software functions and imaged in Z-stacks of 10–12 slices and ≈1 $\mu$m step size in both transmitted light and green channel fluorescence. The grid maps were saved as .lif files for subsequent identification of the transfected cells at the cryo-transmission electron microscope (cryo-TEM).

Data acquisition was performed using a Thermo Fisher Scientific Titan Krios G4 TEM equipped with a Thermo Fisher Scientific Selectris X energy filter and a Thermo Fisher Scientific Falcon 4i direct electron detector. The microscope was operated at 300 keV in zero-loss mode with an energy filter slit width set to 10 eV. To identify the area of interest for data collection, the map acquired on the Leica Thunder was overlaid with the TEM images acquired with MAPS (TFS) software. Tomograms were acquired at underfocus from 4 to 6 microns, with a 33K magnification resulting in a 0.376-nm pixel size at the specimen level, using SerialEM software (Mastronarde, 2005). The collection scheme used was dose-symmetric, covering an angular range from −60° to +60° with 2° increments, starting at 0°. The cumulative electron dose was ~120 e−/Å2. All image stacks were motion-corrected using alignframes IMOD (Kremer et al, 1996) and reconstructed with AreTomo (Zheng et al, 2022).

## Manual image analysis

All manual cell quantifications were performed in Fiji ImageJ using the CellCounter function, processed with Microsoft Excel, and plotted in GraphPad Prism. For manual analysis of Onda-11, U87, and H4 cell morphology, we assigned GFP+ cells to one of the defined morphoclasses and morphotypes. The same was done for Ki67, where GFP+ Ki67+ cells were assigned to one of the three different Ki67 patterns of expression. PVim, Casp3, EdU, L1CAM, A2B5, nestin, GFAP, OCT4, and SOX2 positivity was also calculated using the CellCounter function in Fiji ImageJ. All images were analyzed blindly.

For the time-lapse movies, GFP+ morphoclasses were manually tracked over time and scored for the morphological change in the interphase and mitosis. MST was defined as the distance the nucleus travels during the time step preceding mitosis. Maximum projections and generations of movies were carried out in Fiji ImageJ.

For the neurosphere assay, image analysis was carried out in Fiji ImageJ where the area of the core and the total neurosphere (including the protrusions) was measured with the freehand line tool. The invasion index was calculated by dividing the area of the core by the total area of the neurosphere.

## Automated image analysis

For the machine learning–assisted pipeline for image analysis (Fig 3), we collected a total of 39 microscopy images, out of which we segmented the morphology of 1,362 Onda-11 cells, using CellPose, an artificial neural network for automated cell segmentation. The "cyto2" pretrained model was chosen and retrained for improved Onda-11 cell segmentation. Each cell was labeled through its own image array using Python in a Jupyter Notebook. As a first step, each cell was positioned singularly at the center of a new image array with the dimension of the biggest bounding box and saved as "tiff" file. Subsequently, the following morphological features were extracted: area, perimeter, major and minor axis lengths, and eccentricity. These properties were engineered using the "region-props_table" function from the scikit-image library to compute properties (measurements) out of labeled regions in the image arrays. Eccentricity is a measure of cellular elongation and circularity, where an eccentricity equal to 0 indicates a circle, whereas values between 0 and 1 indicate an ellipse.

To analyze Onda-11 cell protrusions, we modified our previous semi-manual workflow named Progenitors Process Analysis (PPA) (Kalebic et al, 2019) and used to quantify the number of primary and all protrusions, average and maximum protrusion length, branching index (ratio between the total number of protrusion and primary protrusions), and Sholl analysis. The source code of the scripts and helper library is available on an online repository (git platform of HT; KalebicLab/morphoADD3 (github.com)). Additional details on the installation, usage, and implementation of the workflow can be found on that repository.

## Data-driven selection of ADD3

To identify genes that potentially regulate GSC morphology, we used a published tumor atlas of differentially expressed genes in primary GBM tumors (Bhaduri et al, 2020). We intersected this dataset with a list of morphoregulatory genes involved in neuro-development identified in Kalebic et al (2019). This yielded a list of 30 candidate genes. The enrichment of adducins among the 30 genes was calculated through a hypergeometric test with the following parameters: the total number of human protein-coding genes = 19,396 (N), total number of adducins = 3 (n), number of selected genes = 30 (k), and number of hits = 3 (x). We then investigated the expression level of the selected genes (29 of 30 genes as one of the genes, MGEA5, was not analyzed in the datasets mentioned below) in 48 annotated GBM cell lines from Cancer Dependency Map dataset (22Q2 version) (Tsherniak et al, 2017; Behan et al, 2019; Pacini et al, 2021) and the Sanger Cell Model Passports (van der Meer et al, 2019) observing a bimodal distribution from which we identified 18 highly expressed genes (whose basal expression was seemingly generated by the distribution with the higher mean). Subsequently, we derived the depletion fold change of these 18 genes upon CRISPR/Cas9 targeting in 48 GBM cell lines using the same resources. We excluded pan-cancer core-fitness genes (as predicted in Vinceti et al [2021]) and focused our

attention on ADD3 as an important morphoregulator during development (Kalebic et al, 2019), differentially expressed in GBM (Bhaduri et al, 2020) and with a strong and context-specific depletion fold change in GBM cell lines. We then identified Onda-11 as the GBM cell line with the highest dependency on ADD3. U-87 MG was selected as a GBM cell line with low or no ADD3 dependency, whereas H4 was selected as a glioma cell line with mild ADD3 dependency.

### RNA-sequencing and gene expression analyses

During sorting, GFP+ Onda-11 GSCs were collected in lysis buffer containing RNA inhibitors in nuclease-free water. RNA was extracted through SMART-Seq v4 Ultra Low Input RNA Kit for Sequencing (Takara). The libraries were sequenced with NovaSeq 6000 with SP flow cell and the following read configuration: 150 × 10 × 10 × 150. Reads from the same sample, obtained from different sequencing lanes, were aggregated and subjected to adapter trimming using Trim Galore. Processed reads were aligned to the human reference genome (GRCh38) using STAR, and quantification was performed with Salmon. Count data were regularized and log-transformed using the *rld* built-in DESeq2 function, and samples were clustered based on Euclidean distances. Differential expression analysis was performed using DESeq2 using raw counts as input. Differentially expressed genes were identified using a cutoff of absolute $\log_2$ fold change ($\log_2$ FC) ≥ 0.5 and a false discovery rate (FDR) < 0.05. To comprehensively evaluate the outcomes of the differential expression analysis, we employed Cancer Cell Line Encyclopedia (CCLE) profiles—standardized to achieve zero mean and unit variance—across 48 GBM cell lines. We calculated pairwise correlation scores across all genes, considering the upper triangle of this matrix as a null distribution of scores. Pairwise Pearson's correlation scores between ADD3 and DEGs were extracted and compared with the null with a *t* test. The source code of the scripts is available on an online repository (git platform of HT; https://github.com/Raf91/ADD3-project/tree/main).

### Statistical analysis

All statistical analyses were conducted using Prism (GraphPad Software). To test for statistical significance ($P < 0.05$), two-way ANOVA with the Sidak or Bonferroni post hoc tests, the Fisher exact test, and a *t* test were used. For each graph, the number of samples, statistical test, and the *P*-value are noted in the figure legends.

## Data Availability

The gene expression data from this publication have been deposited to the GEO database (https://www.ncbi.nlm.nih.gov/geo/) and assigned the identifier GSE280761.

## Supplementary Information

## Acknowledgements

We are grateful to the services and facilities of HT for the outstanding support provided, notably, N Maghelli and F Casagrande from the National Facility (NF) for Light Imaging, D dalle Nogare from the NF for Data Handling and Analysis, C Peano and the team of the NF for Genomics, P Swuec and the team of the NF for Structural Biology, and A Pallini and the team of the flow cytometry infrastructural unit. We thank C Ossola (Kalebic Lab) for the ADD3 clone. We are thankful to B Soskic (HT) and E Argenzio (HT) for the critical reading of the article, R Galli (HSR) for useful comments, and all members of the Kalebic Lab for helpful discussions. The icons used in the graphical abstract were taken from the online platforms Alamy and BioRender. C Barelli and RM Iannuzzi are Ph.D. students within the European School of Molecular Medicine (SEMM). This work has been supported by funds of HT and the grant from AIRC (MFAG 2022 ID 27157) to N Kalebic.

### Author Contributions

C Barelli: conceptualization, data curation, formal analysis, validation, investigation, visualization, methodology, and writing—original draft, review, and editing.
F Kaluthantrige Don: formal analysis, investigation, visualization, and methodology.
RM Iannuzzi: formal analysis, visualization, and methodology.
S Faletti: formal analysis, investigation, visualization, methodology, and writing—review and editing.
I Bertani: formal analysis, validation, investigation, and methodology.
I Osei: software and methodology.
S Sorrentino: investigation.
G Villa: investigation.
V Sokolova: investigation.
A Campione: resources.
MR Minotti: resources.
GM Sicuri: resources.
R Stefini: resources.
F Iorio: supervision, methodology, and writing—review and editing.
N Kalebic: conceptualization, formal analysis, supervision, funding acquisition, visualization, project administration, and writing—original draft, review, and editing.

### Conflict of Interest Statement

F Iorio receives funds from Open Targets, a public–private initiative involving academia, and from Nerviano Medical Sciences S.r.l and performs consultancy for the Cancer Research Horizons-AstraZeneca Functional Genomics Centre and for Mosaic Therapeutics.

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
