## [Reviewer comments · Life Science Alliance]

Life Science Alliance

Morphoregulatory ADD3 underlies glioblastoma growth and formation of tumor-tumor connections

Carlotta Barelli, Flaminia Kaluthantrige Don, Raffaele Iannuzzi, Stefania Faletti, Ilaria Bertani, Isabella Osei, Simona Sorrentino, Giulia Villa, Viktoria Sokolova, Alberto Campione, Matteo Minotti, Giovanni Sicuri, Roberto Stefini, Francesco Iorio, and Nereo Kalebic

DOI: <https://doi.org/10.26508/lsa.202402823>

Corresponding author(s): Nereo Kalebic, Human Technopole

Review Timeline:

Submission Date:	2024-05-14
Editorial Decision:	2024-07-15
Revision Received:	2024-09-27
Editorial Decision:	2024-10-21
Revision Received:	2024-11-02
Accepted:	2024-11-04

Transaction Report:

July 15, 2024

Re: Life Science Alliance manuscript #LSA-2024-02823-T

Dr. Nereo Kalebic
Human Technopole
Viale Rita Levi Montalcini 1
Milano, MI 20157
Italy

Dear Dr. Kalebic,

Thank you for submitting your manuscript entitled "Morphoregulatory ADD3 underlies glioblastoma growth and formation of tumor-tumor connections" to Life Science Alliance. The manuscript was assessed by expert reviewers, whose comments are appended to this letter. We invite you to submit a revised manuscript addressing the Reviewer comments.

Thank you for this interesting contribution to Life Science Alliance. We are looking forward to receiving your revised manuscript.

Sincerely,

B. MANUSCRIPT ORGANIZATION AND FORMATTING:

Reviewer #1 (Comments to the Authors (Required)):

Morpho regulatory ADD3 underlies glioblastoma growth and formation of tumor-tumor connections

Authors: Barelli et al.

Journal: Life Science Alliance

Summary: In this manuscript, Barelli et al have elucidated a key morpho regulatory role for adducin 3 (ADD3) in glioblastoma stem cells (GSCs). Through a series of elegant experiments involving immunostaining and genetic modelling supplemented with some omics-based data, the authors have identified ADD3 as a key regulator of GSC morphology. They show that it localizes in GSC protrusions and influences the distribution of distinct morphological GSC subtypes favoring an elongated subtype in GSCs. Interestingly, they show that ADD3 expression is important in driving GSC proliferation and its knockout results in decreased GSC proliferation and a modest increase in cell death. It is important to note that the data reported in this manuscript contrasts some published literature in glioblastomas (PMID: 33172155, PMID: 31958485 etc.) which predict a more anti-tumor role for ADD3. These prior studies do not focus on cell-specific roles for ADD3, thus making the findings of Barelli et al important in advancing our understanding of adducins and their role in glioblastomas. The authors also highlight the cell-autonomous effects of ADD3 expression along with its role in mediating GSC self-renewal and treatment resistance in GBMs. While some of their claims are not entirely supported, overall, their findings have strong translational relevance. The manuscript is well written, and the authors have used suitable controls wherever necessary. However, I have a few concerns which if addressed will help clarify the GSC-specific role for ADD3 in driving glioblastomas. If the below listed concerns are addressed preferably through experiments or by discussion, I will be happy to endorse the manuscript for publication as it will be suitable for the journal's broad readership.

Major Comments:

1. The data presented in the manuscript points to a GSC-specific role for ADD3. The authors use 2 cell line (Onda11 and U-87) models grown in serum-free conditions for their experiments. While the authors have used robust stemness markers to define Onda11 and U87 cells that phenocopy GSCs, we do not know what percentage of the entire population of tumor cells resemble GSCs? In other words, under serum free conditions, do the Onda11 and U-87 cell lines transform completely to exhibit stem cell characteristics?
2. To overcome the above-mentioned shortcoming, can the authors perform immunostaining on patient GBM tumor tissues to show ADD3 staining in Glioma stem cells (these can be identified by the stemness markers used by the authors)?
3. As an extension of the previous question, does ADD3 expression in GSCs confer worse prognosis in GBM patients? The authors can use published single-cell RNA sequencing data to define GSC_ADD3_high and GSC_ADD3_low patients based on median expression of ADD3 in GSCs and check if that can be used as a prognostic indicator. This would further emphasize the need for understanding cell-type specific roles for ADD3 and other such cytoskeletal modulators.
4. Do the authors expect a similar role for ADD1 in promoting GBM morphology since it was also identified in their initial screen? If not, what makes ADD3 unique with regard to glioblastomas?

Referee Cross-Comments:

Reviewer 2 shares many of my concerns with some of the conclusions drawn by the authors in the manuscript mainly with regard to how they define GSCs. While I agree with reviewer 2 on using primary GBM tumors and identifying glioma stem cells from the heterogenous tumor tissue, if acquiring fresh GBM tissue proves to be difficult, the authors should validate their initial findings using IHC on patient tissues. Without this, data from a homogenous cell line may weaken the authors' conclusions.

Reviewer #2 (Comments to the Authors (Required)):

The manuscript by Barelli et al., describes the role of adducin-3 (ADD3) in glioma/GBM cells and suggests that ADD3 regulates cell-cell connection/communication and cell proliferation. The intriguing aspect of the study is that the authors propose that cell morphology is functionally linked to cell-cell connection, and by extension cell proliferation.

Major issues

Further, the authors refer to GSC (GBM stem cells) but use GBM cell lines cultured in serum-free medium. My understanding is that GSC are the cancer-initiating cells, and not simply a GBM cell converted into a neurosphere/serum-free sub-line of the parental cells. If the authors wish to use the term GSC, then they need to use patient-derived primary GBM cells which were selected in & maintained in serum-free medium & express the markers they show in Fig S1C. Otherwise, they should refer to these as neurosphere cells. On this point, in Fig S1C, the cell morphology in the CD44 panel, looks very different to all other cells in the FigS1C. Are these a different cell type?

The ADD3 KO or OE cells exhibit changes in cellular protrusions & microtubules. While the disruption ADD3 expression appears to lead to these changes, I question how specific this is to ADD3, given that many other mutations in neural progenitor cells lead to the same/more severe effects in filopodium or invadopodium function, which are likely indirect effects with respect to filopodia or invadopodia function, (<https://doi.org/10.1038/ncb1654>) (doi: 10.1093/neuonc/noy068), unless the authors can show an invadopodium-specific functional effect, e.g. <https://doi.org/10.18632/oncotarget.25045>. The authors should discuss this.

The authors suggest that microtubule communication with surrounding cells regulates cell survival - whether this is due to microtubules or paracrine factors/ extracellular vesicles, is unclear. Can the authors explain further on whether this has been tested?

The suggestion that cell morphology regulates proliferation and other oncogenic function, and that ADD3 has a role in these functions, while interesting, the link is tenuous, as cell morphology in situ & in vitro will be different and depend on both cellular & non-cellular/biophysical factors, so I don't find this argument and the data supportive of this concept.

Minor issues

The manuscript uses terms which are unusual, including 'morphoclass' and 'polar'/'non-polar', in reference to cell morphology. While I think I understand what the authors mean, I have never heard of these terms used in describing cell morphology/biology. Is this terminology unique to this study or can the authors provide a reference which will help explain what these terms mean?

In the introduction, p3, line 71, the authors state that "Here we identified adducin- γ (ADD3), an actin-associated protein known to control bRG morphology and proliferation, as a putative master morphoregulator of GSCs" - please provide references to this backup.

Overview of revision

Figure	Revision	Contents	Reviewers
New figure 1		Morphological heterogeneity of GSCs in primary GBM tissue	1,2
Figure 2	Old Figure 1 with New panels D and E	ADD3 is expressed by GSCs in primary GBM tissue	1
New figure 6	Panel A taken from old fig.5		
	New panels B, C	ADD3 promotes resistance to temozolomide therapy	1
New figures 3-5, 7	Old figures 2-4 and 6, respectively		
Figure S1	New panel J	Expression of stemness markers in Onda 11 grown in serum	1
	New panels K, L	Onda 11 GSC can form clones in stringent conditions	2
	New panel M	Old Figure 1E	1
Figure S3	New panels E and I	Expression of stemness markers in U87 and H4 grown in serum	1
New figure S11		ADD3 promotes resistance to temozolomide therapy	1
Figure S12	Old figure S11		

Response to reviewers

Reviewer #1

Summary: In this manuscript, Barelli et al have elucidated a key morpho regulatory role for adducin 3 (ADD3) in glioblastoma stem cells (GSCs). Through a series of elegant experiments involving immunostaining and genetic modelling supplemented with some omics-based data, the authors have identified ADD3 as a key regulator of GSC morphology. They show that it localizes in GSC protrusions and influences the distribution of distinct morphological GSC subtypes favoring an elongated subtype in GSCs. Interestingly, they show that ADD3 expression is important in driving GSC proliferation and its knockout results in decreased GSC proliferation and a modest increase in cell death. It is important to note that the data reported in this manuscript contrasts some published literature in glioblastomas (PMID: 33172155, PMID: 31958485 etc.) which predict a more anti-tumor role for ADD3. These prior studies do not focus on cell-specific roles for ADD3, thus making the findings of Barelli et al important in advancing our understanding of adducins and their role in glioblastomas. The authors also highlight the cell-autonomous effects of ADD3 expression along with its role in mediating GSC self-renewal and treatment resistance in GBMs. While some of their claims are not entirely supported, overall, their findings have strong translational relevance. The manuscript is well written, and the authors have used suitable controls wherever necessary. However, I have a few concerns which if addressed will help clarify the GSC-specific role for ADD3 in driving glioblastomas. If the below listed concerns are addressed preferably through experiments or by discussion, I will be happy to endorse the manuscript for publication as it will be suitable for the journal's broad readership.

Author's response:

We would like to thank the Reviewer for their positive comments on our manuscript. We have addressed the concerns raised and we hope that by the Reviewer will find the results convincing.

Reviewer's comment:

The data presented in the manuscript points to a GSC-specific role for ADD3. The authors use 2 cell line (Onda11 and U-87) models grown in serum-free conditions for their experiments. While the authors have used robust stemness markers to define Onda11 and U87 cells that phenocopy GSCs, we do not know what percentage of the entire population of tumor cells resemble GSCs? In other words, under serum free conditions, do the Onda11 and U-87 cell lines transform completely to exhibit stem cell characteristics?

Author's response:

We have now analysed the expression of stemness markers in Onda 11 grown under serum conditions and included it in Fig. S1J. This revealed that while Onda 11 grown in serum have no or very low expression of some stemness markers (such as SOX2, OCT4) they exhibited expression for other markers (GFAP 40%, NES 90%). In contrast when grown in serum-free condition Onda 11 exhibited high expression of

all markers (SOX2, 75%; OCT4 85%; GFAP 80%, NES 100%). This shows that under the serum-free conditions all stem cell markers are upregulated and at least 75% of cells are GSCs, which we report in the relative figure legend. We have also quantified the expression of stemness markers in U87 and H4 cells grown in serum: Fig. S3E (U87) and Fig. S3I (H4).

Reviewer's comment:

To overcome the above-mentioned shortcoming, can the authors perform immunostaining on patient GBM tumor tissues to show ADD3 staining in Glioma stem cells (these can be identified by the stemness markers used by the authors)?

Author's response:

We thank the Reviewer for this important comment. We have visualised GSCs in primary GBM samples by immunofluorescence for Nestin and SOX2 (New Figure 1). This revealed that GSCs are morphologically heterogenous in patient samples, similarly to what we observed in 2D GSC culture. These data are now added to the Results (new chapter entitled "Glioblastoma stem cells (GSCs) exhibit morphological heterogeneity similar to the one observed in neural progenitors during cortical development"). Subsequently, we detected that ADD3 is present in primary GBM samples and it is expressed by 75.2% of GSCs (Fig.2D-E and Results I.140-144).

Reviewer's comment:

As an extension of the previous question, does ADD3 expression in GSCs confer worse prognosis in GBM patients? The authors can use published single-cell RNA sequencing data to define GSC_ADD3_high and GSC_ADD3_low patients based on median expression of ADD3 in GSCs and check if that can be used as a prognostic indicator. This would further emphasize the need for understanding cell-type specific roles for ADD3 and other such cytoskeletal modulators.

Author's response:

Finding good quality single-cell datasets with prognosis indicators proved to be difficult. Thus, to understand whether ADD3 influences patients' prognosis, we tested whether ADD3 promotes chemoresistance to temozolomide (TMZ). We included our results in Fig.6 B, C and Fig. S11 and Results I.342-359. This revealed that ADD3 overexpression results in increased resistance to temozolomide therapy, as we have detected a selected survival of ADD3 over expressing cells. These data suggest that ADD3 likely plays an important role in patients' prognosis.

Reviewer's comment:

Do the authors expect a similar role for ADD1 in promoting GBM morphology since it was also identified in their initial screen? If not, what makes ADD3 unique with regard to glioblastomas?

Author's response:

The key reason for the choice of ADD3 over the other two isoforms of the adducins lies in its already shown role in regulating the morphology and proliferation of neural stem cells in the developing brain (Kalebic et al., 2019). Furthermore, ADD3 has

been linked to GBM by several groups with quite contradictory results, whereas ADD1 had very poor association with cancers and to our knowledge no link to brain tumors (now described in the Results 1.130-138). Given that adducins were shown to operate as heterodimers in which ADD1 is an obligatory partner, it is valid to ask how might ADD3 exert ADD1-independent effects. Variants in ADD3 have been strongly associated to hereditary cerebral palsy, whereas no human mutation in ADD1 was related to this disease, nor does any of the KO mouse models of ADD1 exhibit the associated symptoms. Hence the example of cerebral palsy might be similar to GBM. This could mean that there are specific biochemical roles of ADD3 mediated either through interactions with other proteins or through potential homodimerization. We included some of these thoughts in the Discussion (chapter “ADD3 as a key morphoregulator in GBM”, 1.529-541).

Reviewer's comment:

Referee Cross-Comments:

Reviewer 2 shares many of my concerns with some of the conclusions drawn by the authors in the manuscript mainly with regard to how they define GSCs. While I agree with reviewer 2 on using primary GBM tumors and identifying glioma stem cells from the heterogenous tumor tissue, if acquiring fresh GBM tissue proves to be difficult, the authors should validate their initial findings using IHC on patient tissues. Without this, data from a homogenous cell line may weaken the authors' conclusions.

Author's response:

We thank the Reviewer for their support. We have established a clinical collaboration and have obtained fresh GBM samples that we used to provide analyses presented in Figures 1 and 2D, E, as mentioned above.

Reviewer #2

The manuscript by Barelli et al., describes the role of adducin-3 (ADD3) in glioma/GBM cells and suggests that ADD3 regulates cell-cell connection/communication and cell proliferation. The intriguing aspect of the study is that the authors propose that cell morphology is functionally linked to cell-cell connection, and by extension cell proliferation.

Reviewer's comment:

Further, the authors refer to GSC (GBM stem cells) but use GBM cell lines cultured in serum-free medium. My understanding is that GSC are the cancer-initiating cells, and not simply a GBM cell converted into a neurosphere/serum-free sub-line of the parental cells. If the authors wish to use the term GSC, then they need to use patient-derived primary GBM cells which were selected in & maintained in serum-free medium & express the markers they show in Fig S1C. Otherwise, they should refer to these as neurosphere cells. On this point, in Fig S1C, the cell morphology in the CD44 panel, looks very different to all other cells in the FigS1C. Are these a different cell type?

Author's response:

We would like to thank the Reviewer for their positive comments on our manuscript. The Reviewer raises two concerns and in the revised version we provide the following data in our support, as follows:

As to generation and culturing of GBM stem cells (GSCs):

- (1) We obtained patient samples of GBM and identified GSCs (using SOX2 and Nestin markers) on histological sections (Figure 1). We confirmed that 75% of those primary GBM cells also express ADD3 (Figure 2D, E).
- (2) To specifically examine the stemness of Onda 11 GSCs we have now performed functional clonogenic assays in methylcellulose which revealed that Onda 11 GSCs can form clones in stringent conditions (Figure S1 K, L). These data together with our previous analysis of stemness markers in Onda 11 GSCs (SOX2, 75%; OCT4 85%; GFAP 80%, NES 100%, CD44 90%; Figure S1D-I) strongly suggest that those cells are indeed stem cells.

As for CD44 image (Figure S1C), it labels 90% of Onda 11 GSCs (Figure S1I) so it does not identify a specific cell population, rather it labels the cell membrane and thus has a different appearance from the other nuclear and intermediate filaments markers.

Reviewer's comment:

The ADD3 KO or OE cells exhibit changes in cellular protrusions & microtubes. While the disruption ADD3 expression appears to lead to these changes, I question how specific this is to ADD3, given that many other mutations in neural progenitor cells lead to the same/more severe effects in filopodium or invadopodium function, which are likely indirect effects with respect to filopodia or invadopodia function, (<https://doi.org/10.1038/ncb1654>) (doi: 10.1093/neuonc/noy068), unless the authors can show an invadopodium-specific functional effect, e.g. <https://doi.org/10.18632/oncotarget.25045>. The authors should discuss this.

The authors suggest that microtubule communication with surrounding cells regulates cell survival - whether this is due to microtubules or paracrine factors/ extracellular vesicles, is unclear. Can the authors explain further on whether this has been tested?

The suggestion that cell morphology regulates proliferation and other oncogenic function, and that ADD3 has a role in these functions, while interesting, the link is tenuous, as cell morphology in situ & in vitro will be different and depend on both cellular & non-cellular/biophysical factors, so I don't find this argument and the data supportive of this concept.

Author's response:

The Reviewer raises 4 different points.

As to the specificity of ADD3, we have added a new section in the Discussion (in the chapter "ADD3 as a key morphoregulator in GBM", l.529-541) where we discuss potentially specific roles of ADD3 with respect to other members of the adducin family. While we agree that other mutations in neural progenitor cells might lead to the same effects, in our study we show that ADD3 controls the abundance of tumor-tumor connections and mediates the resistance to chemotherapy, which strongly suggests important and potentially specific roles in brain tumors. Nevertheless, we fully agree that other neurodevelopmentally-related proteins might exert similar functions in GBM and we suggest this in l.570-571. We argue that identification and mechanistic characterization of other potential molecular targets with a neurodevelopmental role could be helpful in future diagnostic and therapeutic approaches in brain cancers. Finally, effects from various targets are likely integrated with other signalling pathways and in this context we thank the Reviewer for mentioning the Daniel et al., 2018; which we references in the section of the discussion "Morphology as a new layer of GBM heterogeneity" (l.443).

As to the specificity of the type of protrusions induced by ADD3, we have shown that ADD3 regulates (1) cell morphology by promoting elongation of the cell body (Fig. 3I-M), (2) cell branching through the growth of primary and all cell protrusions (Fig. 3D-H) and (3) connectivity of GSCs (Fig. 7A-C). We do not see an invadopodium-specific effect as when we cultured Onda 11 GSC neurospheres in matrigel we did not detect a difference in ECM invasion between control and ADD3 OE (Fig S10). We thank the Reviewer for mentioning the reference Petropoulos et al., 2018, which, together with (Ratliff et al., 2023; Venkataramani et al., 2022) we include in the Discussion (chapter "Cell-cell connections link GSC morphology with proliferation, chemoresistance and survival", l.480). This together suggest that a different population of GBM cells, the one which lacks connections to other GBM cells, is most likely the main driver of brain tumor invasion. Finally, as we reported in the Discussion "the question remains if ADD3 directly induces new protrusions by remodeling actin in the membrane cytoskeleton or whether it stabilizes existing protrusions by connecting actin filaments to the plasma membrane." (l.522-525). We now expand the section on actin capping by discussing also how the filopodia formation underlies neurite outgrowth and referencing Dent et al., 2007, as commented by the Reviewer (l.529-530).

As to paracrine factors/extracellular vesicles, we thank the Reviewer for this comment. Whereas cell survival in cancer can be controlled by various signalling modalities including paracrine, autocrine signalling, direct cell-cell signalling, and extracellular vesicles, in this study we focused on the effects of cell-cell communication as we have shown that ADD3 controls to ability to form cell-cell connections. Interestingly, in the literature TTCs have been associated to chemoresistance and GBM growth. We have here provided evidence suggesting that ADD3-induced TTCs affect GSC proliferation, chemoresistance and survival. However, we agree that cell death can not only be mediated by the loss of these cell-cell signals but also through paracrine, autocrine signalling and exchange of extracellular vesicles. We have included these important points in the discussion (I.504-507).

As to the difference in cell morphology *in situ* and *in vitro*, we have now examined patient GBM samples and analysed the morphological heterogeneity of GSCs (new Figure 1 and Results I.93-110). Interestingly, this showed that GSCs in primary GBM tissue have a great morphological heterogeneity and that multiple similarities between 2D and 3D morphotypes can be observed. While we agree that cell morphology is highly influenced by complex environmental and mechanical factors that are lacking in 2D, and that various features of cell morphology will be different between the two conditions, our data do suggest that the morphological heterogeneity observed in GBM samples was broadly recapitulated in our 2D GSC model systems, which in turn implies that basic morphological nature is a cell-intrinsic property. This has now been discussed in the Discussion section “Morphology as a new layer of GBM heterogeneity” (I.436-447).

Reviewer’s comment:

The manuscript uses terms which are unusual, including ‘morphoclass’ and ‘polar’/‘non-polar’, in reference to cell morphology. While I think I understand what the authors mean, I have never heard of these terms used in describing cell morphology/biology. Is this terminology unique to this study or can the authors provide a reference which will help explain what these terms mean?

Author’s response:

We have now better explained these new terms in the results section when they appear for the first time. Whereas *morphotype* is accepted to define a subtype of cells that share the same morphological features (see for example Kalebic and Huttner, 2020), with the term *morphoclass* we refer to a family of morphotypes with the same principal features (now defined in Results, I.164). We have also defined all the 4 morphoclasses (containing term polar/non-polar) as follows: Non-polar cells are polygonal cells without any type of protrusion (suggesting no morphological polarization), flat polar cells are characterized by a big and flat cell body and have some protrusions, circular multipolar cells are small rounded cells with many short protrusions and lastly, elongated cells have a long and thin cell body with one or more long and thin protrusions (I.165-168).

Reviewer’s comment:

In the introduction, p3, line 71, the authors state that "Here we identified adducin-γ (ADD3), an actin-associated protein known to control bRG morphology and proliferation, as a putative master morphoregulator of GSCs" - please provide references to this backup.

Author's response:

We have now added the reference showing that ADD3 is an actin-associated protein (Kiang and Leung, 2018) and the second one showing that it regulates bRG morphology and proliferation (Kalebic et al., 2019) (l.80-81).

October 21, 2024

RE: Life Science Alliance Manuscript #LSA-2024-02823-TR

Dr. Nereo Kalebic
Human Technopole
Viale Rita Levi Montalcini 1
Milano, MI 20157
Italy

Dear Dr. Kalebic,

Thank you for submitting your revised manuscript entitled "Morphoregulatory ADD3 underlies glioblastoma growth and formation of tumor-tumor connections". We would be happy to publish your paper in Life Science Alliance pending final revisions necessary to meet our formatting guidelines.

- please be sure that the authorship listing and order is correct
- please consult our manuscript preparation guidelines <https://www.life-science-alliance.org/manuscript-prep> and make sure your manuscript sections are in the correct order
- please upload your table files as editable doc or excel files
- please add a separate figure legend section (including your main, supplementary, and video legends) to the main manuscript text
- please upload your supplementary figures as single files
- please add a figure callout for Figure S3 A-C; E-G; and S3I

Figure Check:

- please add scale bars to all microscopy images; Figure 2H, Figure 3C, Figure S3, Figure S4E, Figure S5B&D, Figure 6A&B, Figure S7A&D

LSA now encourages authors to provide a 30-60 second video where the study is briefly explained. We will use these videos on social media to promote the published paper and the presenting author (for examples, see <https://docs.google.com/document/d/1-UWCfbE4pGcDdcgzcmiuJl2XMBJnxKYeqRvLLrLSo8s/edit?usp=sharing>). Corresponding or first-authors are welcome to submit the video. Please submit only one video per manuscript. The video can be emailed to contact@life-science-alliance.org

A. FINAL FILES:

B. MANUSCRIPT ORGANIZATION AND FORMATTING:

Sincerely,

Reviewer #1 (Comments to the Authors (Required)):

I am very satisfied with the revised data/manuscript and would recommend accepting it.

Reviewer #2 (Comments to the Authors (Required)):

The authors have improved the manuscript with additional text & further explanations. A minor issue that remains is that all micrograph image panels, including Figs S6 & S7 should show scale bars. Please check all panels & include a scale bar for each.

Response to the Editor and the Reviewers:

We thank the editor and the reviewers for having accepted our revised version of the study and for their useful comments which have now been addressed.

Reviewers' comments:**Reviewer #1 (Comments to the Authors (Required)):**

I am very satisfied with the revised data/manuscript and would recommend accepting it.

Authors' response:

We thank the Reviewer for their kind comment.

Reviewer #2 (Comments to the Authors (Required)):

The authors have improved the manuscript with additional text & further explanations. A minor issue that remains is that all micrograph image panels, including Figs S6 & S7 should show scale bars. Please check all panels & include a scale bar for each.

Authors' response:

We thank the Reviewer for their useful comment. We have now added scale bars in Fig. S6 and 7 and all the other figures where scale bars were missing.

Editor's comments:

-please be sure that the authorship listing and order is correct

We confirm that the authorship listing and order is correct.

-please consult our manuscript preparation guidelines <https://www.life-science-alliance.org/manuscript-prep> and make sure your manuscript sections are in the correct order

We have now put all of the sections in the right order.

-please upload your table files as editable doc or excel files

We have now uploaded the "Table S1. Reagents and Tools table" as a doc file.

-please add a separate figure legend section (including your main, supplementary, and video legends) to the main manuscript text

We have also added the Figure legend, Supplemental figure legends and Supplemental video section to the main manuscript file.

-please upload your supplementary figures as single files

We have now uploaded both main and supplementary figures as single files.

-please add a figure callout for Figure S3 A-C; E-G; and S3I

We have now added a figure callout for all panels of the Figure S3 and accordingly rephrased several sentences in the "ADD3 is sufficient and required to control the number of protrusions and elongation of GSCs" (lines 208-218).

Figure Check:

-please add scale bars to all microscopy images; Figure 2H, Figure 3C, Figure S3, Figure S4E, Figure S5B&D, Figure 6A&B, Figure S7A&D
We have now added scale bars to all the figure panels.

November 4, 2024

RE: Life Science Alliance Manuscript #LSA-2024-02823-TRR

Dr. Nereo Kalebic
Human Technopole
Viale Rita Levi Montalcini 1
Milano, MI 20157
Italy

Dear Dr. Kalebic,

Thank you for submitting your Research Article entitled "Morphoregulatory ADD3 underlies glioblastoma growth and formation of tumor-tumor connections". It is a pleasure to let you know that your manuscript is now accepted for publication in Life Science Alliance. Congratulations on this interesting work.

DISTRIBUTION OF MATERIALS:

Again, congratulations on a very nice paper. I hope you found the review process to be constructive and are pleased with how the manuscript was handled editorially. We look forward to future exciting submissions from your lab.

Sincerely,
